

# RH and O₃ concentration as two prerequisites for sulfate formation

Yanhua Fang[1][#] and Chunxiang Ye[1][#], Junxia Wang[1], Yusheng Wu[1], Min Hu[1], Lin Weili[2], Fanfan Xu[1], Tong Zhu[1][*]

[1]BIC-ESAT and SKL-ESPC, College of Environmental Sciences and Engineering, Peking University, Beijing, 100871, China
[2]College of Life and Environmental Sciences, Minzu University of China, Beijing 100081, China

[#]These authors contributed equally to the paper.

[*]*Correspondence to*: Tong Zhu (tzhu@pku.edu.cn)

**Abstract.** Sulfate formation mechanisms have been discussed extensively but are still disputed. In this work, a year-long
particulate matter (PM$_{2.5}$) sampling campaign was conducted together with measurements of gaseous pollutant concentrations
and meteorological parameters in Beijing, China, from March 2012 to February 2013. The sulfur oxidation ratio (SOR), an
indicator of secondary sulfate formation, displayed a clear summer peak and winter valley, even though no obvious seasonal
variations in sulfate mass concentration were observed. A rapid rise in the SOR was found at a RH threshold of ~45% or an
O$_3$ concentration threshold of ~35 ppb, suggesting that RH and O$_3$ concentrations were two prerequisites for rapid sulfate
formation, which likely occured via multiphase reactions. H$_2$O$_2$ oxidation was proposed to be the major route of sulfate
formation, since the O$_3$ oxidation route has previously been shown to be unimportant. The seasonal variations in sulfate
formation could be accounted for by variations in the RH and O$_3$ prerequisites. For example, over the year-long study, the
fastest SO$_2$-to-sulfate conversion occurred in summer, which was associated with the highest values of both O$_3$ concentration
and RH. The SOR also displayed variations with pollution levels, i.e., the SOR increased as pollution evolved in all seasons.
Such variations were primarily associated with a transition from the slow gas phase formation of sulfate to rapid multiphase
reactions, since RH increased as pollution evolved. In addition, the self-catalytic nature of sulfate formation (i.e., increasing
aerosol water content with simultaneous increases in sulfate mass concentrations and RH) also contributed to variations among
the pollution scenarios. Overall, our observations validated the two prerequisites for fast sulfate formation and revealed the
seasonal and pollution level variations in sulfate formation in Beijing. H$_2$O$_2$ oxidation was the major route of sulfate formation,
although reactions involving transition metal ions (TMIs) and NO$_2$ might have also competed.

## 1 Introduction

Beijing, the capital of China, suffers from serious air pollution due to its rapid economic growth and urbanisation (Hu et al.,
2015a). The chemical composition and sources of fine particulate matter (PM$_{2.5}$) in Beijing have been studied extensively (Han
et al., 2015; Lv et al., 2016; Zhang et al., 2013; Zheng et al., 2005). Secondary components, especially sulfate, nitrate, and





ammonium (SNA), are the main contributors to PM$_{2.5}$ (Huang et al., 2014). On the most severely polluted days, SNA account for more than half of total PM$_{2.5}$ mass concentrations and play a more important role than on clean days (Quan et al., 2014; Wang et al., 2014b; Zheng et al., 2015b).

The kinetics and mechanisms of the formation of sulfate, a major component of SNA, are complex and remain unclear (Ervens,
2015; Harris et al., 2013; Warneck, 2018). For example, two key questions concerning sulfate formation are: (1) exactly how do various parameters influence sulfate formation, and (2) how do multiple formation routes compete and contribute together to sulfate formation under ambient conditions. In general, sulfate is produced from SO$_2$ via gas phase oxidation reactions involving the hydroxide radical (OH) and Criegee intermediates (Gleason et al., 1987; Sarwar et al., 2014; Vereecken et al., 2012), heterogeneous reactions (mainly on dust aerosols), and multiphase transformations with O$_3$, H$_2$O$_2$, or O$_2$ (catalysed by
transition metal ions (TMIs) (i.e., TMIs + O$_2$) and NO$_2$ (NO$_2$ + O$_2$)) as liquid phase oxidants, which occur mainly in clouds but also in aerosol droplets near the ground (Zhu et al., 2011).

Due to the major role of multiphase transformations, sulfate production is presumed to be self-catalysed, i.e., the formation of hydrophilic sulfate aerosols under high relative humidity (RH) conditions results in an increase in aerosol water content (AWC), which results in greater particle volume for further multiphase reactions (Cheng et al., 2016; Pan et al., 2009; Xu et al., 2017).
Analyses of the correlation of sulfate formation with RH and AWC have been conducted to test this hypothesis, using the concept of the sulfur oxidation ratio (SOR), defined as the molar ratio of sulfate to total sulfur (= sulfate + SO$_2$). It is used to indicate the magnitude of the secondary formation of sulfate and expressed as (Wang et al., 2005):

$$\text{SOR} = \frac{n_{\text{SO}_4^{2-}}}{n_{\text{SO}_4^{2-}} + n_{\text{SO}_2}}, \tag{Eq. 1}$$

where $n_{\text{SO}_4^{2-}}$ and $n_{\text{SO}_2}$ represent the molar concentrations of sulfate and SO$_2$, respectively.
Sun et al. (2014; 2013) found positive correlations between the SOR and RH, and observed rapid increases in SORs at elevated RH levels. Xu et al. (2017) found positive correlations of the SOR with both RH and AWC. Multiphase transformation routes, including O$_3$ oxidation, TMIs + O$_2$, and NO$_2$ + O$_2$ routes, are pH-sensitive and suppressed at low pH (Seinfeld and Pandis, 2006). Sulfate production raises the acidity of aerosols and therefore the multiphase transformations of sulfate are presumed to be self-constrained (Cheng et al., 2016). For example, a significant contribution from the O$_3$ oxidation route can only be
expected under alkaline conditions (e.g., sea-salt), otherwise, O$_3$ oxidation is only a minor pathway for sulfate formation (Alexander et al., 2005; Sievering et al., 2004). How the self-constraining nature of sulfate formation influences the relative significance of the TMI + O$_2$ and NO$_2$ + O$_2$ routes is still under debate. Cheng et al. (2016) proposed that the NO$_2$ + O$_2$ route is important during severe haze under neutral conditions (He et al., 2018; Wang et al., 2016). Guo et al. (2017) suggested that aerosols are acidic in Beijing (except for during the limited cases of dust or sea-salt events), casting doubt on the importance
of the NO$_2$ + O$_2$ route in sulfate formation (Liu et al., 2017a). According to laboratory-based Raman spectroscopy studies, sulfate can be produced via the aqueous oxidation of SO$_2$ by NO$_2$ + O$_2$, with an SO$_2$ reactive uptake coefficient of 10$^{-5}$, which represents an atmospherically relevant value (Yu et al., 2018), whereas others have suggested that this route is of minor importance in the atmosphere (Li et al., 2018; Zhao et al., 2018). In addition, Xie et al. (2015) proposed that NO$_2$ could enhance



the formation of sulfate in certain cases, for example, in biomass burning plumes or dust storms (He et al., 2014). Evaluation of the contribution of TMI + $O_2$ reactions appears to be more complex since it depends on aerosol acidity, solubility, oxidation state, and the synergistic effects of different TMIs (Deguillaume et al., 2005; Warneck, 2018).

The compensating effects among AWC, aerosol acidity, and the concentrations of precursors and catalysts show that the

kinetics and mechanisms of sulfate formation are highly complex. It can be inferred that there is competition between the various routes, with dependences on atmospheric conditions (e.g., seasonal and pollution level variations) likely, but this has not received much research attention previously. Here, daily $PM_{2.5}$ samples were collected in Beijing from March 2012 to February 2013 and their chemical composition was analysed. The main parameters that influenced sulfate formation (i.e., RH, $O_3$ concentration, TMIs, etc.) were determined. This valuable dataset enabled us to explore: (1) the specific role of each

influencing factor in sulfate formation, and (2) how multiple sulfate formation routes compete in different seasons and under various pollution scenarios.

## 2 Measurements and methodology

### 2.1 Measurements

#### 2.1.1 Measurement stations

The two measurement stations are shown in Figure 1. The PKU station (116.30 °E, 39.99 °N) is about 20 m above ground level on the campus of Peking University, Beijing, China (Liang et al., 2017). Daily $PM_{2.5}$ samples were collected using a four-channel sampler (TH-16A; Wuhan Tianhong Instruments, China) at a flow rate of 16.7 L min$^{-1}$ from 1 March 2012 to 28 February 2013. The gaseous pollutants $SO_2$, $NO_x$, and $O_3$ were measured with a pulsed fluorescence $SO_2$ analyser (Model 43i TLE; Thermo Fisher Scientific, Waltham, MA, USA), chemiluminescence $NO–NO_2–NO_x$ analyser (Model 42i TL; Thermo

Fisher Scientific), and an ultraviolet photometric $O_3$ analyser (Model 49i; Thermo Fisher Scientific), respectively. Temperature and RH were also monitored (MSO; Met One Instruments, Grants Pass, OR, USA). Solar radiation data were obtained from the Beijing Meteorological Observatory Station.

#### 2.1.2 Filter sampling and analysis

Each $PM_{2.5}$ sample set consisted of one quartz filter (47 mm; Whatman QM/A, Maidstone, England) and three Teflon filters

(47 mm; pore size = 2 μm; Whatman PTFE). The quartz filters were baked for 5.5 h at 550 °C before use. The Teflon filters were weighed in a weighing room before and after sampling using a delta range balance (0.01 mg/0.1 mg precision; AX105; Mettler Toledo, Switzerland). To minimise contamination, all Teflon filters were placed in a super clean room (temperature = 22 ± 1 °C; RH = 40 ± 2%) for 24 h before being weighed. After sampling, all filters were stored at −20°C prior to analysis.

Water soluble cations ($Na^+$, $NH_4^+$, $K^+$, $Mg^{2+}$, and $Ca^{2+}$) and anions ($SO_4^{2-}$, $NO_3^-$, $Cl^-$, and $F^-$) were measured using ion

chromatography (ICS-2500 and ICS-2000; DIONEX, USA). Trace elements (Na, Mg, Al, Ca, Ti, Cr, Mn, Fe, Co, Ni, Cu, Zn,



Se, Mo, Cd, Ba, Tl, Pb, Th, and U) were analysed by inductively coupled plasma–mass spectrometry (ICP–MS, X-Series; Thermo Fisher Scientific). Organic carbon (OC) and elemental carbon (EC) were measured using a thermal/optical carbon analyser (RT-4; Sunset Laboratory Inc., Tigard, OR, USA). The procedure for the measurement of water soluble Fe has been described in detail in a previous study (Xu et al., 2018).

## 2.2 Estimation of the mass concentrations of PM$_{2.5}$ components

The chemical components of PM$_{2.5}$ were divided into seven categories according to their source type: sulfate, nitrate, ammonium, organic matter (OM), EC, minerals, and trace element oxides (TEOs). The mass concentrations of OM, minerals, and TEOs were calculated from OC, Al, and trace element concentrations, respectively. The details of this method are provided in the supplementary information (SI). For minerals, validation of the method using only Al to represent all minerals is shown in Fig. S2. TEOs mostly originated from anthropogenic sources (Fig. S3).

## 2.3 Quality assurance and quality control

The PM$_{2.5}$ sampling instruments were cleaned and calibrated every 2–3 months. Before the daily filter replacement, filter plates were scrubbed with degreasing cotton that had been immersed in dichloromethane. For water-soluble ions, OC/EC, and trace element measurements, standard solutions were analysed before each series of measurements. The $R^2$ values of the calibration curves were all > 0.999. For water-soluble ion measurements, beakers, tweezers, and vials were cleaned with deionised water (18.2 MΩ; Milli-Q, USA) three times before use. Certified reference standards (National Institute of Metrology, China) were used for calibration. For OC/EC measurements, tweezers and scissors were scrubbed with degreasing cotton immersed in dichloromethane for every filter. Total organic carbon (TOC) was calculated based on calibration with external standard solutions. For trace element measurements, containers and tweezers were cleaned three times with nitric acid before use, and the analysis of a certified reference standard (NIST SRM-2783) was used to verify accuracy. The recovery of all measured trace elements fell within ±20% of their certified values. For gaseous pollutants and meteorological parameters, all instruments were maintained and calibrated weekly based on manufacturers' protocols.

## 3 Results and discussion

### 3.1 General description

The annual and seasonal mean (± one standard deviation (SD)) concentrations of PM$_{2.5}$ and its seven major components are summarised in Table 1. The annual mean PM$_{2.5}$ concentration was 84.1 (±63.1) μg m$^{-3}$, which is more than two times greater than the Chinese National Ambient Air Standard of 35 μg m$^{-3}$. On 145 of the 318 (46%) measurement days, daily mean PM$_{2.5}$ concentrations were above the Chinese National Ambient Air Standard 24 h mean concentration of 75 μg m$^{-3}$. Time series of PM$_{2.5}$ concentrations and its seven major components are shown in Fig. 2. Seasonal variations in PM$_{2.5}$ loading are obvious, with spring and winter peaks and summer and autumn valleys. OM and EC concentrations display common seasonal variations,



with a plateau from mid-October to mid-February and a valley in summer (Fig. 2), which resembles the variations in $PM_{2.5}$, $K^+$, and $Cl^-$ (Figs. 2 and 4). Further investigation revealed that primary organic matter (POM) was the main component of OM (Fig. S4). These results suggest that OM, EC, $K^+$, and $Cl^-$ may have originated from a common primary source. There are clear temperature dependencies among OM, EC, $K^+$, and $Cl^-$ concentrations, which are elevated at low temperatures (i.e., −5 to

15 ℃) (Fig. S5). Coal combustion for heating purposes was initially considered to be their common source. However, a tracer of coal combustion emissions, Se (Ranville et al., 2010), exhibits different seasonal variations, with a concentration valley from spring to mid-October (Fig. 4). The Se concentrations had a strong dependence on temperature, where high concentrations appeared only when the temperature was below 0 ℃ (Fig. S5). The period with high Se concentrations coincided with the heating season in Beijing in 2013 (Fig. 4). Therefore, we further conclude that OM, EC, $K^+$, and $Cl^-$ did originate from

anthropogenic sources, but that coal combustion was not their sole primary source. Other anthropogenic sources should be considered, such as biomass burning, to account for the emissions of OM, EC, $K^+$, and $Cl^-$ during the heating season (Liang et al., 2017; Tan et al., 2016; Zhang et al., 2015b). Overall, anthropogenic sources contributed much more to $PM_{2.5}$ mass loading from mid-October to mid-February than in summer and early autumn. Changes in anthropogenic sources thus dominated the seasonal variations in $PM_{2.5}$ and might have implications for secondary chemistry, such as sulfate formation. SNA accounted

for more than one-third of $PM_{2.5}$ annually and showed similar seasonal variations to that of $PM_{2.5}$ (Fig. 2), with the notable exception that sulfate became the highest contributor to $PM_{2.5}$ (~25%) in summer (Fig. 3). The summer peak in sulfate could be accounted for by fast secondary formation, as will be discussed later.

The seasonal variations in minerals also indicated an important contribution of dust events to $PM_{2.5}$ loading during spring, which is a well-known phenomenon (Zhang et al., 2003; Zhuang et al., 2001). TEOs display no obvious seasonal variations

(Fig. 2). Based on the high enrichment factors of most TEOs, they were mostly derived from anthropogenic sources (Fig. S3). For example, the sources of Zn and Pb include industrial combustion and waste incineration (Hu et al., 2015b). There was no clear explanation for the lack of seasonal variations in TEOs.

On an annual basis, the seven major components accounted for over 80% of $PM_{2.5}$ (Fig. 3). The diversity of the seasonal variations in $PM_{2.5}$ and its major components found in our study imply that there were seasonal variations in both the primary

sources and secondary formation of $PM_{2.5}$.

### 3.2 Influence of various parameters on sulfate formation

To further explore the parameters that influenced sulfate formation, SORs were plotted against RH and the concentrations of $O_3$, $NO_2$, and Fe (total Fe, including both water soluble and water insoluble Fe), which is a major tracer of transition metals (Figs. 5 and 6).

As shown in Fig.5a, an RH threshold of ~45% was critical for efficient $SO_2$ oxidation (i.e., a high SOR). Such a threshold effect was thought to be reasonable given that aerosol water content (AWC) increases sharply at a given RH threshold, at which the aerosol undergoes a phase transition from a (semi-)solid particle to a droplet (Pan et al., 2009; Russell and Ming,





2002). Our observation of a daily average RH threshold of ~45% is in line with previous reports of 40–50% (Liu et al., 2015; Quan et al., 2015; Xu et al., 2017; Yang et al., 2015; Zheng et al., 2015b), but is slightly lower than the *in situ* phase transition threshold RH of 50–60% previously observed in Beijing (Liu et al., 2017b). Correlation analysis of SOR and RH (or AWC) has often been conducted in previous studies. For example, Wang et al. (2005) found a weak positive correlation of SORs with

RH ($R = 0.38$), while Sun et al. (2006) found a strong positive correlation ($R = 0.96$). However, the analysis in the present work and those of a few previous studies revealed that the relationship between the SOR and RH is nonlinear (Sun et al., 2013; Sun et al., 2014; Zheng et al., 2015b). In fact, the RH threshold suggests that high RH (or AWC) is a prerequisite for fast sulfate formation via multiphase reactions, which are known to account for the majority of sulfate accumulation.

From the large scattering of data points around the fit line in Fig. 5a, it might be inferred that RH was not the only prerequisite

for fast $SO_2$-to-sulfate conversion. As shown in Fig. 5b, a significant increase in the SOR was also observed at an $O_3$ concentration threshold of ~35 ppb. High $O_3$ concentrations (i.e., > 35 ppb) were accompanied by high SOR values of ~0.4 (right-hand side of Fig. 5b). Correlation analyses of SORs with $O_3$ have been conducted but inconsistent results were reported. Wang et al. (2005) found a weak positive correlation between SORs and $O_3$ ($R = 0.47$) for continuous observations in Beijing during 2001–2003. However, Liu et al (2015) found a weak negative correlation between SORs and $O_3$ ($R = -0.53$, $p = 0.01$)

during a haze episode in September 2011. Zhang et al. (2018) found no correlation between SORs and $O_3$ during winter haze days in 2015. Quan et al. (2015) found that the SOR decreased with $O_3$ when $O_3$ concentrations were lower than 15 ppb, but increased with $O_3$ when $O_3$ concentrations were higher than 15 ppb, for observations made during autumn and winter 2012. In the present study, our observations revealed that the relationship between the SOR and $O_3$ concentration, like RH, was nonlinear and that a high $O_3$ concentration was another prerequisite for fast sulfate formation. Such a conclusion was a surprise

first, since $O_3$ oxidation was not thought to be a major route for $SO_2$-to-sulfate conversion (Sievering et al., 2004). However, as a primary precursor to OH radicals and $H_2O_2$ (via $HO_2$), $O_3$ is also considered to be a proxy for atmospheric oxidative capacity (Lelieveld et al., 2016; Lu et al., 2017). High $O_3$ concentrations (e.g., > 35 ppb) correspond to a strong oxidative capacity, which favors multiphase sulfate formation and thus a high SOR, whereas low $O_3$ concentrations suggest a lack of available oxidants for multiphase $SO_2$-to-sulfate conversion and thus a low SOR. In addition, the simultaneous occurrence of

low SORs and low $O_3$ concentrations had a secondary cause. Low $O_3$ concentrations in the Beijing urban area were often due to the titration of $O_3$ by NO (Li et al., 2016), which accumulated together with $SO_2$ (Fig. S6). The accumulation of $SO_2$, which "diluted" the SOR (Eq. 1), was thus naturally accompanied by the titration of $O_3$. The L-shaped dependence of the SOR on several other primary pollutants, such as EC, NO, and Se (Fig. S7), further confirmed this secondary cause. Therefore, the accumulation of primary pollutants might also help to explain the low SOR values of ~0.1 on the left-hand side of Fig. 5b, in

addition to the lack of available oxidants for multiphase $SO_2$-to-sulfate conversion.

The large scattering of data points around the fit line in Fig. 5b suggests that $O_3$ concentration, like RH, was not the only prerequisite for fast $SO_2$-to-sulfate conversion. The dependence of the SOR on RH was separated into low (< 35 ppb) and high (> 35 ppb) $O_3$ groups (solid black circles and solid blue circles, respectively, in Fig. 5a). SOR values above the fit line are found mostly for the high $O_3$ group. After the dependency of the SOR on $O_3$ concentration was separated into low (< 45%)



and high ($> 45\%$) RH groups (solid black circles and solid blue circles, respectively, in Fig. 5b), a similar pattern was found for the high RH group. In other words, fast multiphase $SO_2$-to-sulfate conversion could only occur when both $O_3$ and RH exceeded their respective thresholds simultaneously. As the $O_3$ oxidation route can be excluded in a heavily polluted area with high aerosol acidity (Shen et al., 2012; Sievering et al., 2004), such as Beijing, the two prerequisites suggest that $H_2O_2$ oxidation

might be the major sulfate formation route. Such a conclusion is supported by a recent report of model underestimations of $H_2O_2$ concentrations in the Beijing area (Ye et al., 2018).

No clear dependence of the SOR on concentrations of Fe or $NO_2$ was found (Figs. 6a and 6b). Possible reasons and implications of this result will be discussed in the following section. There is currently no available direct measurement for aerosol pH (Rindelaub et al., 2016), and it is not possible to perform a proxy calculation for estimation of aerosol acidity from the data

collected in this work, hence discussion of the self-constrained nature of sulfate formation is beyond the scope of our study.

### 3.3 Implications for sulfate formation mechanisms

Our observations of the factors that influence sulfate formation have implications for sulfate formation routes and the variations among mechanisms. Thresholds in RH and $O_3$ concentrations were found to be critical to the SOR, suggesting that AWC and atmospheric oxidative capacity were two prerequisites for fast multiphase $SO_2$-to-sulfate conversion. These dependences imply

that $H_2O_2$ oxidation played a major role in sulfate formation, as the $O_3$ oxidation route could be excluded due to the strong aerosol acidity in environments such as Beijing, as was stated above. The plot of SORs against Fe, the dominant transition metal species, shows no clear dependence (Figs. 6a and S8). However, such a pattern did not exclude TMIs + $O_2$ as a major route for sulfate formation. Possible explanations for the lack of an Fe dependence are: (1) Fe was accompanied by low RH and $O_3$ concentrations, which are the two main prerequisites for fast $SO_2$-to-sulfate conversion, and (2) Fe acted as a catalyst

and thus its concentration might not be directly proportional to that of the reaction product, sulfate. The relationships of Fe with RH and $O_3$ were examined and the results suggest that TMIs were not related to RH or $O_3$ concentrations (Figs. 6c and 6d), which excluded the first possibility. The plot of SORs against $NO_2$ shows no clear dependence either (Fig. 6b). However, such a pattern does not exclude $NO_2$-based reactions as major routes of sulfate formation. Several laboratory studies excluded $NO_2$ as a direct oxidant in $SO_2$-to-sulfate conversion. For example, Zhao et al. (2018) tested the oxidation of $SO_2$ by $NO_2$ in

an $N_2$ atmosphere and concluded that $NO_2$ is not an important oxidant, since $NO_2$ was more likely to undergo disproportionation (Li et al., 2018). However, Yu et al. (2018) further explored this reaction, and found that the reaction rate was 2–3 orders of magnitude greater in an $O_2 + N_2$ atmosphere, indicating potentially important roles of $NO_2 + O_2$ oxidation in sulfate formation (He et al., 2014; Ma et al., 2018). As with Fe, if $NO_2$ acted as a catalyst, its concentration might not be directly proportional to that of sulfate. Previously, aerosol surface area and concentrations of Fe, Mn, and $NO_2$ were used in model evaluations of

catalytic sulfate formation in the boundary layer (Wang et al., 2014a; Zheng et al., 2015a). However, our observations suggest that a careful reassessment of such calculations is required. In addition, model calculations have often suggested important contributions of in-cloud processes to sulfate accumulation near the ground (Barth et al., 2000), although few observational constraints are available for confirmation of these model results (Harris et al., 2014; Shen et al., 2012). The $O_3$ concentration



and RH prerequisites found in the present work might indicate a major role of *in situ* sulfate formation in the boundary layer, via multiphase reactions with $H_2O_2$ as the main oxidant, rather than in-cloud processes and intrusion from the free troposphere. As the two prerequisites showed strong seasonal and pollution level variations over the measurement year, the SOR exhibited corresponding variations. As shown in Fig. 7, SORs display clear seasonal variations, with the highest value ($\pm 1$ SD) of 0.46

($\pm 0.22$) in summer, followed by spring (0.23 $\pm 0.14$), autumn (0.18 $\pm 0.15$), and winter (0.09 $\pm 0.05$). The highest SOR (i.e., fastest $SO_2$-to-sulfate conversion rate) was found in summer, which is not surprising because the ambient conditions in summer were conducive $SO_2$-to-sulfate conversion (Wang et al., 2005). RH and $O_3$ concentrations in summer were not only the highest in the year, but on average were also both higher than their thresholds of 45% and 35 ppb, respectively, which was unique among the four seasons. In summer, the median and mean ($\pm 1$ SD) RH levels were 57.4% and 57.6 ($\pm 13.6$)%, respectively,

and the median and mean $O_3$ concentrations were 46.9 ppb and 46.0 ($\pm 18.3$) ppb. It should be noted that the median and mean $SO_2$ concentrations were 2.6 and 4.0 ($\pm 3.7$) ppb, respectively, which were the lowest in the year. Despite the low concentrations of $SO_2$, there were considerable sulfate concentrations (Figs. 2 and 7), which can be accounted for by fast $SO_2$-to-sulfate conversion. As discussed above, $H_2O_2$ oxidation could have been a major route of sulfate formation. Direct measurements of aqueous $H_2O_2$ were not conducted in this study, but the $H_2O_2$ concentrations in Beijing reported by Fu (2014) were found to

be highest in summer and elevated from March to November 2013. Although the rapid accumulation of secondary sulfate during winter haze days in Beijing has been widely reported (Wang et al., 2014b; Zheng et al., 2015b), the lowest SOR was observed during winter in the present study (Fig. 7a), which is consistent with previous observations (Wang et al., 2005). On winter haze days, RH values of up to 73.6% and $PM_{2.5}$ mass loadings of up to 375.3 µg m$^{-3}$ were observed. Therefore, AWC was not the limiting factor in $SO_2$-to-sulfate conversion (Figs. 7b and 7e). The $SO_2$-to-sulfate conversion rate in winter could

have been limited by the reduced atmospheric oxidative capacity (Fig. 7c), which was a result of both high emissions of the primary pollutant NO (Fig. S9) and low solar radiation levels (Fig. 7f). Sulfate concentrations in winter were comparable to those in summer, which might have been driven by high $SO_2$ concentrations in winter (Fig. 7d), despite slow $SO_2$-to-sulfate conversion. The lower boundary layer height in winter relative to other seasons would also have encouraged the accumulation of both $PM_{2.5}$ and its components, including sulfate (Gao et al., 2015; Zhang et al., 2015a). The SORs in spring and autumn

were comparable and moderate, possibly representing a transition in conditions between summer and winter.

For each season, four pollution scenarios were classified according to $PM_{2.5}$ level. The lowest 25%, 25–50%, 50–75%, and highest 25% of pollution levels were defined as "clean", "formation", "evolution", and "further evolution", respectively. The relative contributions of seven major components of $PM_{2.5}$ among the four pollution scenarios are shown in Fig. 8. In all four seasons, the relative contribution of SNA increased with $PM_{2.5}$ loading. This phenomenon has been reported in previous studies,

but data availability was limited in autumn (Xu et al., 2017) and winter (Zheng et al., 2015b). The SOR increased consistently in all four seasons as pollution accumulated, where both the highest value and strongest variability were observed in summer (Fig. 9a). Although $SO_2$ should have reduced the SOR (Eq. 1), concurrent increases in primary $SO_2$ and SORs were observed (Figs. 9a and 9b), indicating a significant increase in the $SO_2$-to-sulfate conversion rate with $PM_{2.5}$ loading, which offset the "dilution" effect (Eq. 1). Such variations in sulfate formation with pollution levels can be accounted for by the corresponding



variations in both $O_3$ concentrations and RH (Figs. 9c and 9d). In all four seasons, RH increased consistently as pollution accumulated (Fig. 9d). $O_3$ concentrations decreased consistently as pollution evolved in all of the seasons except for summer (Fig. 9c). The increase in $SO_2$-to-sulfate conversion with $PM_{2.5}$ loading can be attributed to the self-catalytic nature of the multiphase formation of sulfate, i.e., both RH and $PM_{2.5}$ increased continuously with the accumulation of $PM_{2.5}$, resulting in a

rapid rise in AWC and providing greater reaction volume for further sulfate formation. Therefore, the increases in RH and $PM_{2.5}$ could have compensated for the weak oxidative capacity, resulting in fast sulfate formation as pollution evolved. Particularly in summer, not only did both RH and $O_3$ concentrations increase as pollution evolved, but both RH and $O_3$ concentrations were generally above their respective thresholds at all pollution levels (dashed lines in Figs. 9c and 9d). This explains our observations of both the highest value and strongest dependence on pollution level for SORs in summer.

**4 Conclusions**

In this study, the annual mean concentration of $PM_{2.5}$ in Beijing during 2012–2013 was 84.1 ($\pm$ 63.1) $\mu g\ m^{-3}$, with clear seasonal and pollution level variations in its chemical components, highlighting the contribution of SNA formation to the accumulation of $PM_{2.5}$ in all seasons. RH and $O_3$ concentrations were identified as two prerequisites for fast $SO_2$-to-sulfate conversion. RH above a threshold of ~45% greatly accelerated the conversion rate. A similar effect was also found for $O_3$ at a

concentration threshold of ~35 ppb. Such dependences have interesting implications. First, they indicate a major role of the $H_2O_2$ route in sulfate formation, which might further indicate a major role of *in situ* sulfate production in the boundary layer, rather than in-cloud processes and intrusion from the free troposphere. Second, the observed dependences were also able to account for the seasonal and pollution level variations in $SO_2$-to-sulfate conversion rates. Both the highest value and strongest variability of SOR were observed in summer, which might be attributed to the highest values of $O_3$ concentrations and RH in

summer. $SO_2$-to-sulfate conversion accelerated as pollution accumulated, which was primarily attributed to a shift from gas phase oxidation to the multiphase oxidation route, which is self-catalytic in nature. The increase in RH was able to offset the low oxidative capacity under heavily polluted conditions, and resulted in increasingly fast $SO_2$-to-sulfate conversion as pollution accumulated. While our simultaneous observations of the SOR and concentrations of Fe and $NO_2$ could not exclude TMIs + $O_2$ and $NO_2$-based reactions, a reassessment of the relationships between sulfate formation, aerosol surface area and

the concentrations of Fe and $NO_2$ is necessary.

**Author contributions**

TZ designed the study. YHF, CXY, and TZ prepared the manuscript with input from all co-authors. YHF and JXW collected and weighed the $PM_{2.5}$ filter samples and carried out the analysis of the components of $PM_{2.5}$. FFX carried out the measurement of water soluble Fe. YSW and MH provided the data for gaseous pollutants, temperature, and RH. WLL provided the solar

radiation data.





## Competing interests.

The authors declare that they have no conflict of interest.

## Acknowledgements

This work was supported by the National Natural Science Foundation Committee of China (91544000, 41121004, and
91744206). We also thank Dr. Robert Woodward-Massey for his kind help in English writing.

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



**Figures and Tables**

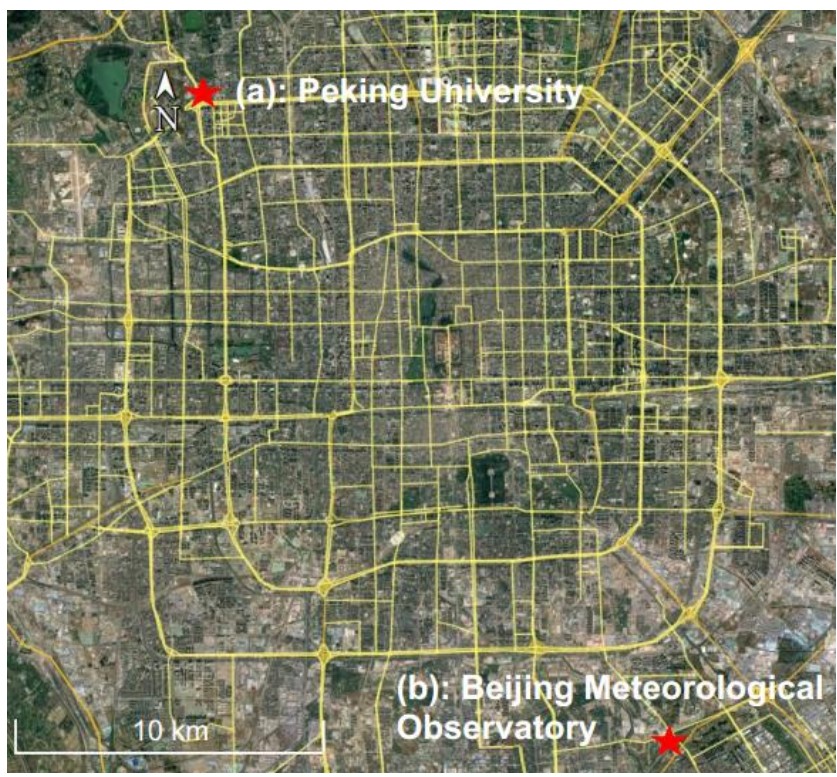

**Figure 1.** Sample sites in this study (red stars): (a) Peking University and (b) Beijing Meteorological Observatory.




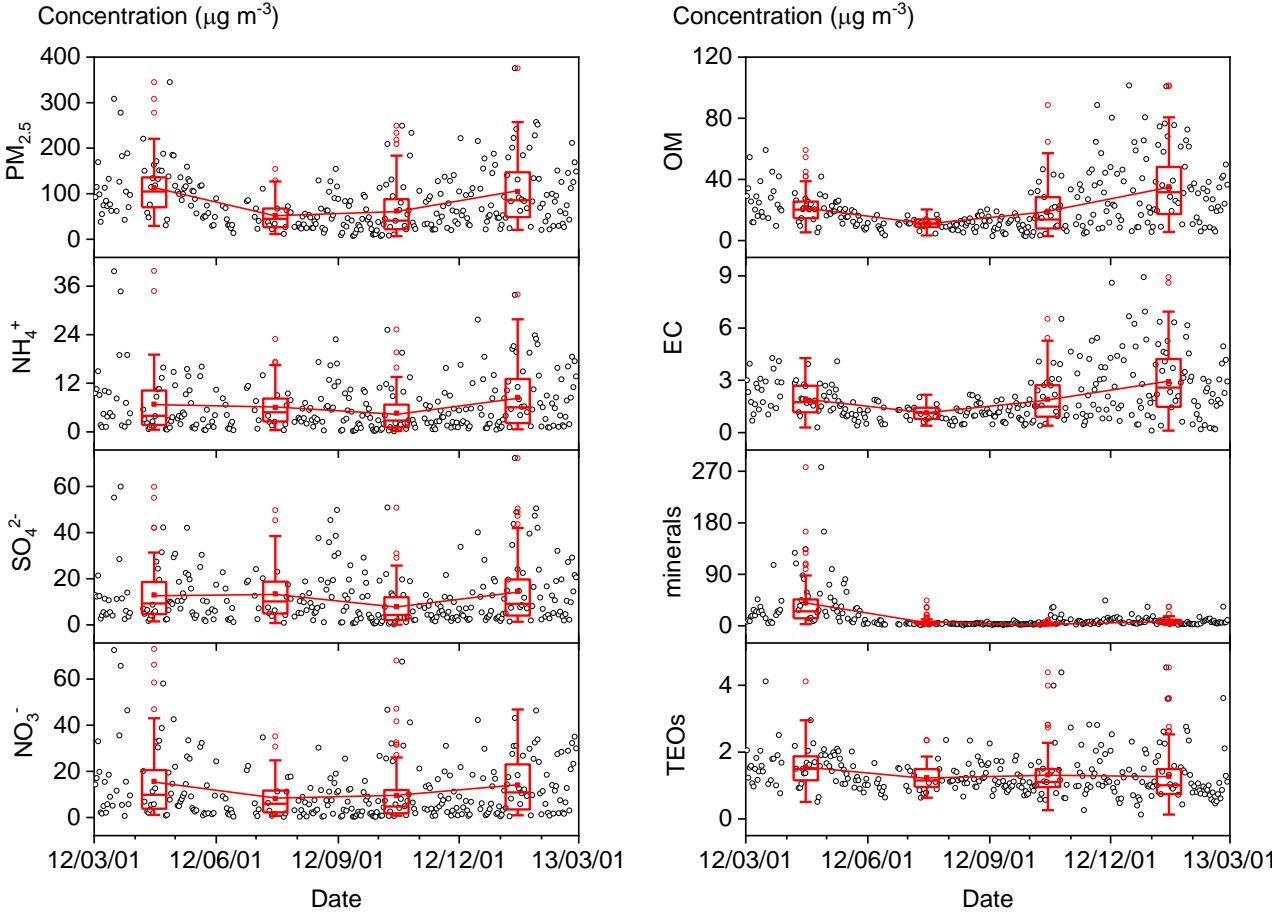

**Figure 2.** Time series of fine particulate matter (PM$_{2.5}$) concentrations and its seven major components from March 1 2012 to February 28 2013 (open black circles). The boxes represent, from top to bottom, the 75[th], 50[th], and 25[th] percentiles for each season. The whiskers, solid red squares, and open red circles represent 1.5 times the interquartile range (IQR), seasonal mean values, and outlier data points, respectively.





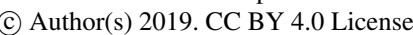

**Figure 3.** Seasonal variations in PM$_{2.5}$ and its seven major components from March 1 2012 to February 28 2013.





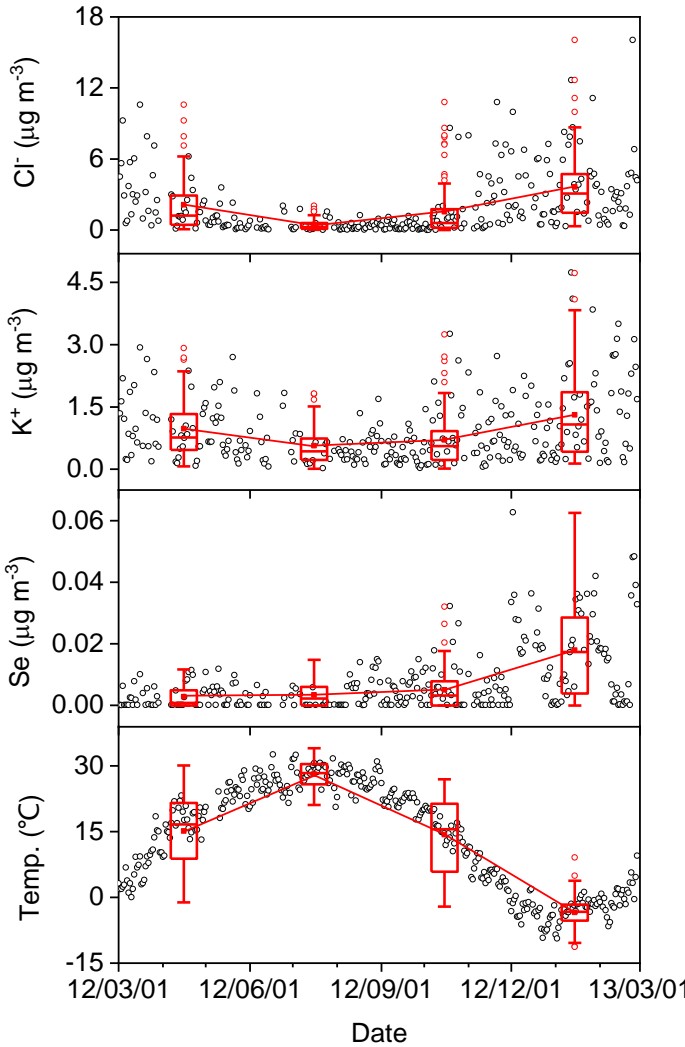

**Figure 4.** Time series of Cl⁻, K⁺, Se, and temperature from March 1 2012 to February 28 2013 (open black circles). The boxes represent, from top to bottom, the 75th, 50th, and 25th percentiles for each season. The whiskers, solid red squares, and open red circles represent 1.5 times the IQR, seasonal mean values, and outlier data points, respectively.





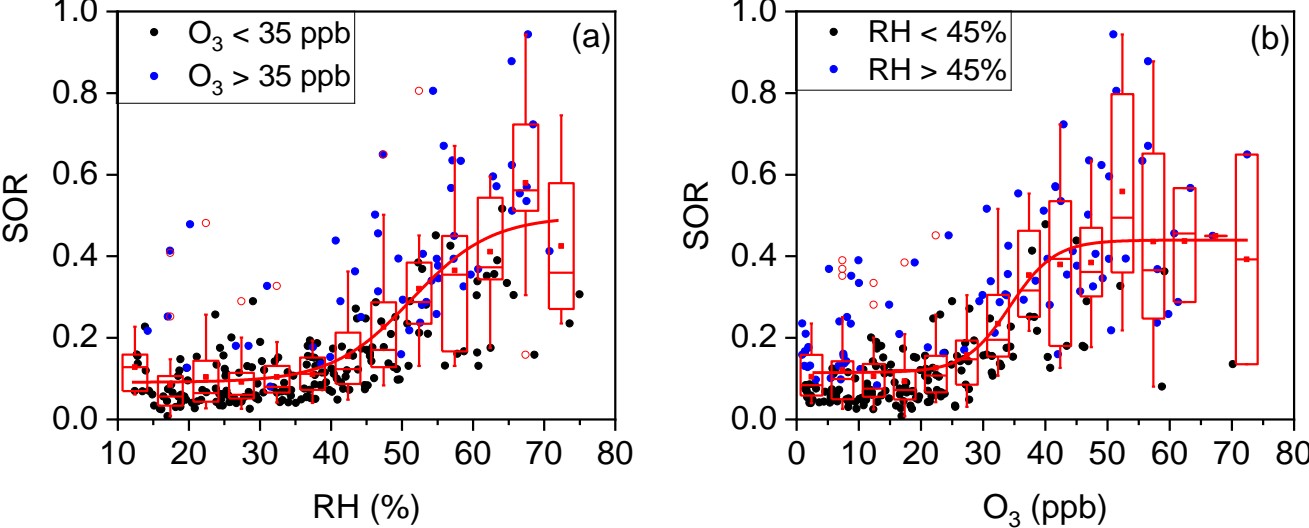

**Figure 5.** (a) Plot of the sulfur oxidation ratio (SOR) against relative humidity (RH) grouped by $O_3$ concentration. The solid blue circles represent $O_3 > 35$ ppb and the solid black circles represent $O_3 < 35$ ppb. (b) Plot of the SOR against $O_3$ grouped by RH. The solid blue circles represent RH > 45% and the solid black circles represent RH < 45%. The boxes represent, from top to bottom, the 75th, 50th, and 25th percentiles in each bin ($\Delta$RH = 5%; $\Delta O_3$ = 5 ppb). The whiskers, solid red squares, and open red circles represent 1.5 times the IQR, seasonal mean values, and outlier data points, respectively. The red lines are best fits to mean values based on a sigmoid function (dose-response). Data for days with rain or snow were excluded from these plots.



**Figure 6.** Plots of SORs against (a) Fe and (b) NO₂. Plots of Fe against (c) RH and (d) O₃. Data for days with rain or snow were excluded from these plots.




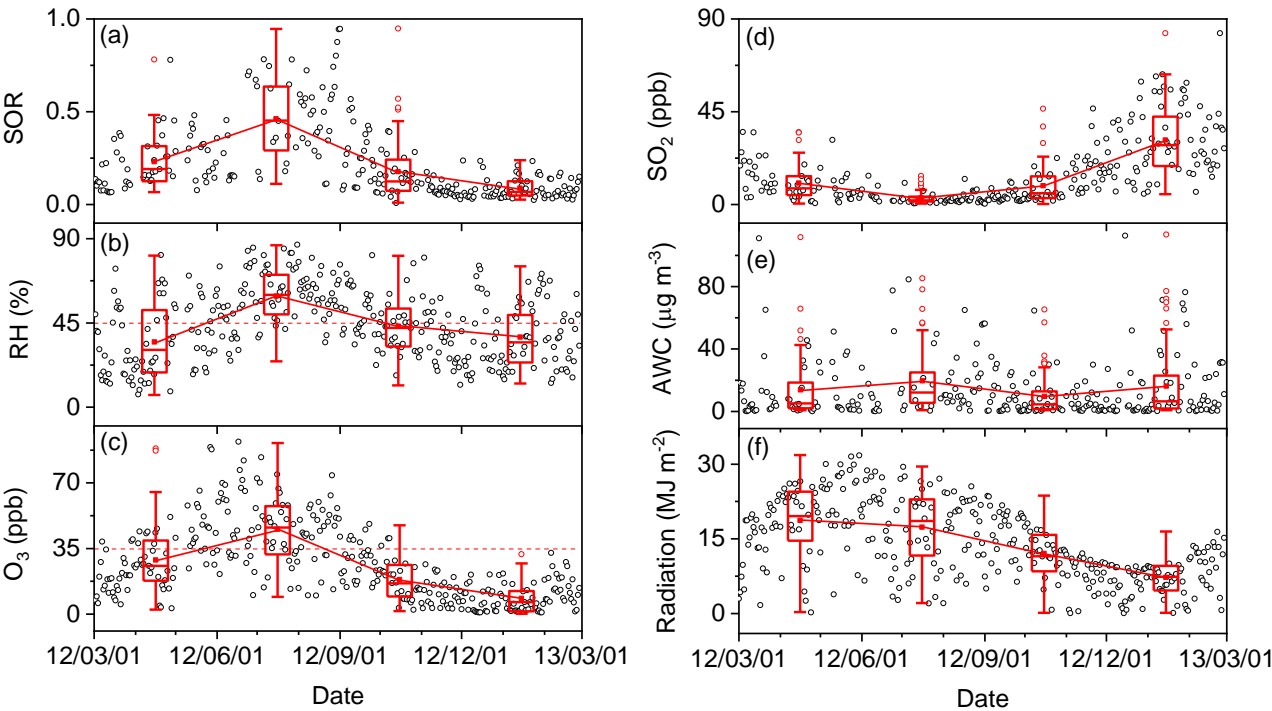

**Figure 7.** Time series of (a) SORs, (b) RH, (c) $O_3$, (d) $SO_2$, (e) aerosol water content (AWC), and (f) solar radiation from March 1 2012 to February 28 2013 (open black circles). The boxes represent, from top to bottom, the 75th, 50th, and 25th percentiles for each season. The whiskers, solid red squares, and open red circles represent 1.5 times the IQR, seasonal mean values, and outlier data points, respectively. The horizontal dashed lines in panels (b) and (c) represent thresholds of RH = 45% and $O_3$ = 35 ppb, respectively.



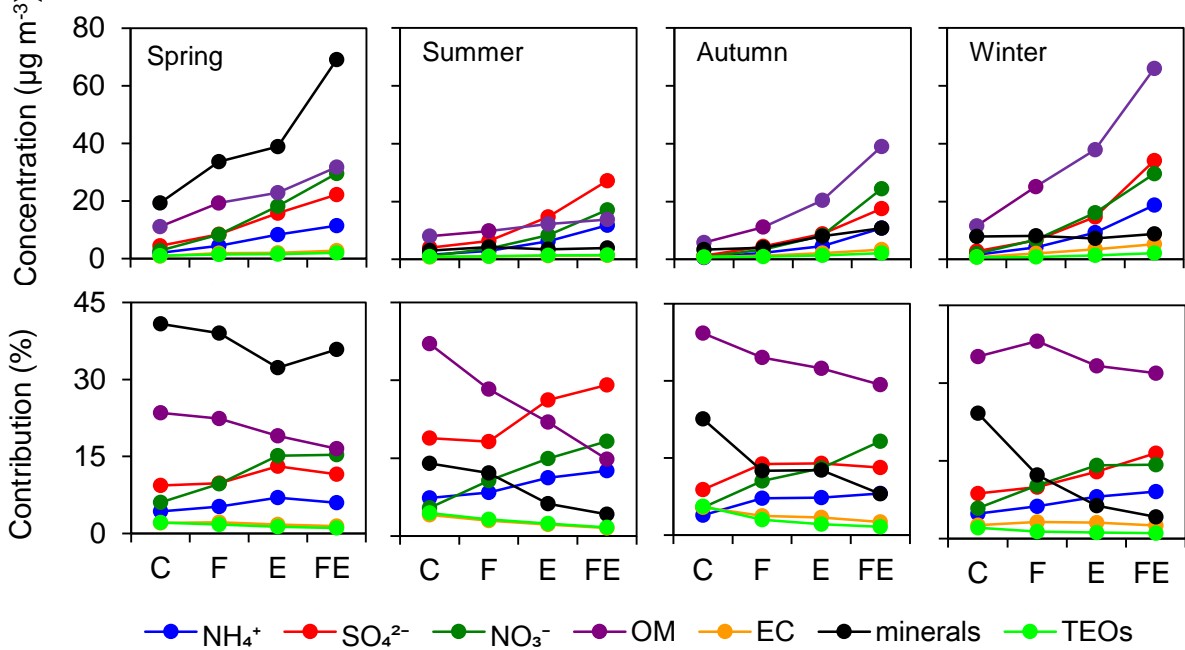

**Figure 8.** Variations in the mean concentrations (upper panels) and contributions (lower panels) of the seven major components of PM$_{2.5}$ with pollution levels in each season. C, clean; F, formation; E, evolution; FE, further evolution.





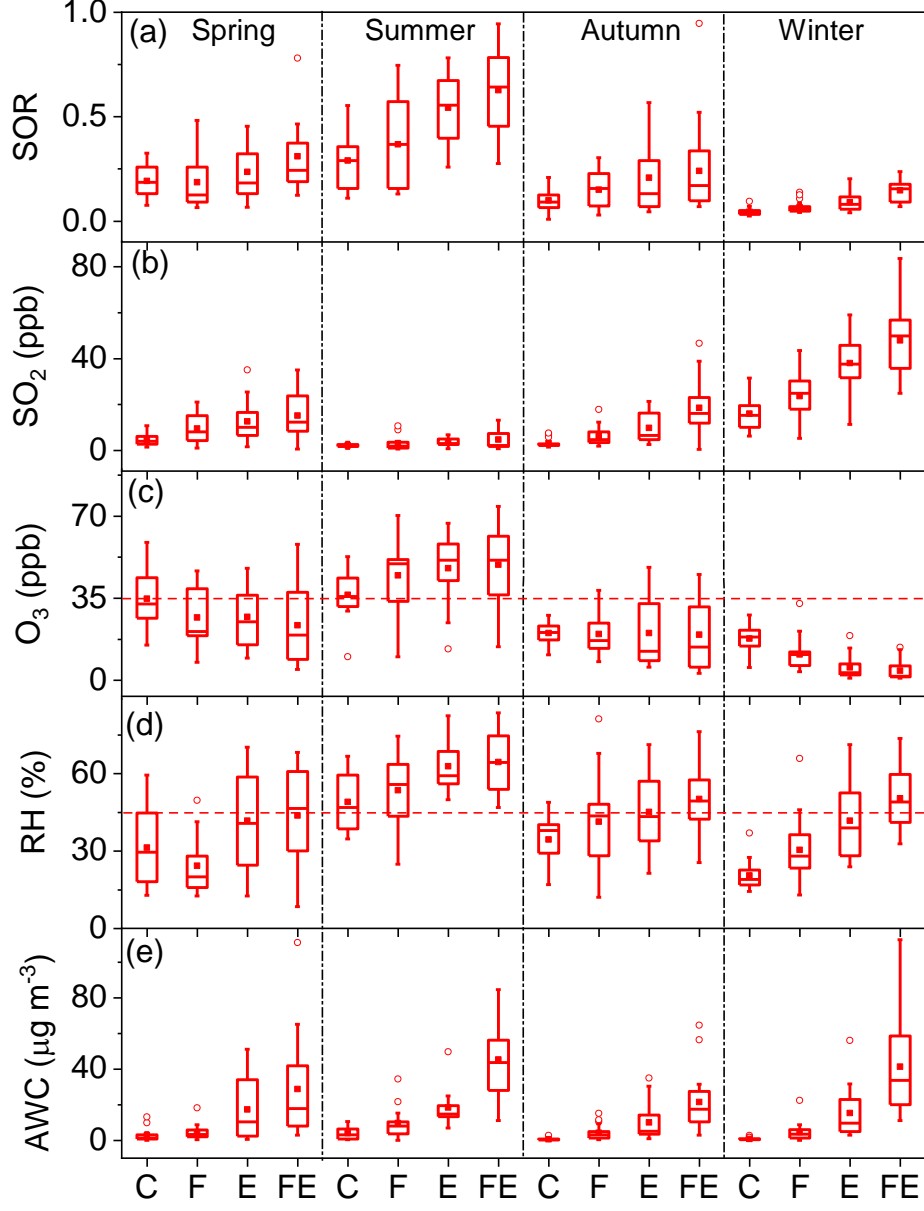

**Figure 9.** Variations in (a) SORs, (b) $SO_2$, (c) $O_3$, (d) RH, and (e) AWC with pollution levels in each season. C, clean; F, formation; E, evolution; FE, further evolution. The boxes represent, from top to bottom, the 75th, 50th, and 25th percentiles for each pollution level. The whiskers, solid red squares, and open red circles represent 1.5 times the IQR, seasonal mean values, and outlier data points, respectively. The horizontal dashed lines in panels (c) and (d) represent thresholds of $O_3$ = 35 ppb and RH = 45%, respectively.



**Table 1.** Annual and seasonal mean concentrations (µg m$^{-3}$, $\pm 1$ standard deviation) of PM$_{2.5}$ and its seven major components.

| Component | Annual | Spring | Summer | Autumn | Winter |
|---|---|---|---|---|---|
| **PM$_{2.5}$** | 84.1 $\pm$63.1 | 113.1 $\pm$62.0 | 52.7 $\pm$32.6 | 60.0 $\pm$51.3 | 105.0 $\pm$71.7 |
| **NH$_4^+$** | 6.4 $\pm$6.4 | 6.7 $\pm$7.3 | 5.9 $\pm$5.0 | 4.5 $\pm$4.8 | 8.4 $\pm$7.4 |
| **SO$_4^{2-}$** | 12.0 $\pm$12.2 | 12.9 $\pm$12.4 | 13.3 $\pm$11.5 | 7.9 $\pm$8.7 | 14.5 $\pm$14.4 |
| **NO$_3^-$** | 11.5 $\pm$12.6 | 15.0 $\pm$16.0 | 7.6 $\pm$8.0 | 9.0 $\pm$11.8 | 13.6 $\pm$12.1 |
| **OM** | 22.7 $\pm$18.1 | 21.5 $\pm$10.5 | 11.1 $\pm$3.8 | 19.2 $\pm$16.1 | 35.2 $\pm$23.4 |
| **minerals** | 14.7 $\pm$27.0 | 40.7 $\pm$45.0 | 3.7 $\pm$1.6 | 6.5 $\pm$7.0 | 8.0 $\pm$5.6 |
| **TEOs** | 1.3 $\pm$0.7 | 1.5 $\pm$0.6 | 1.2 $\pm$0.4 | 1.3 $\pm$0.7 | 1.3 $\pm$0.8 |
| **EC** | 2.1 $\pm$1.5 | 1.9 $\pm$1.0 | 1.1 $\pm$0.5 | 1.9 $\pm$1.3 | 2.9 $\pm$2.0 |