# Peer review of "RH and O3 concentration as two prerequisites for sulfate formation"

_Atmospheric Chemistry and Physics, 2019_

## Referee Comment (RC1) · Anonymous Referee #2 · 29 Apr 2019

The paper deals with the mass concentration and chemical composition of PM2.5 in Beijing during 1 year from filter samples and its correlation with pollution classes (clear days, slight, light, medium and heavy pollution). Most of the paper is devoted to the two prerequisites for sulfate formation based discussion. This is certainly a positive feature of the paper. Although the article has a clear logical structure, I strongly recommend to make the text more concise, to clarify statements, and to delete redundancies. Most importantly, in the absence of data on hydrogen peroxide, all speculation seems weak. The main idea of the article is still in the cognition of previous studies, and no more innovative conclusions have been put forward. In a word, this article is full of paradoxical conclusions and cannot provide a powerful help to the scientific community. Therefore, I don't recommend the publication in ACP journal in current status.

[Figure]

(1) The author name should be Weili Lin. (2) "threshold of RH and ozone" Where is this statement coming from? Is it a definition/estimate of the authors? If the threshold changed with different locations and seasons? What is the effect of these thresholds? (3) Redundancy: Page 1 line 15-16 and line 24-25. Line 13-14 and Line 17-18. (4) Section 2.1.2. Please add the steps of weighing after sampling. (5) Page 4, line 27. Should be annual standard. (6) Page 5, line 2. The method to calculate POM should be introduced in previous section. (7) Overall, section 3.1 is not necessary, because it has nothing to do with the main idea. If this section is deleted in the main article, it will not affect the presentation of the article. For example, the authors described the measurements of ions, organics and metal. However, ions except SNA, organics and metals except Fe didn't help the discussion of your topic. Therefore, the method and results section should to be streamlined. (8) Section 3.2. I strongly recommend the authors discussing the relationship between sulfate and RH/ozone in different seasons. The threshold should be changed with seasons. (9) Page 7, line 12-16 repeats the previous statement. (10) Page 7, line 14. What is the atmospheric oxidative capacity? From your statement, does ozone concentration correspond to this? Is it correct? Do you have some references to support your opinion? The authors should clarify this question because the same definition is also used in Page 9, line 20. (11) Page 7, Line 23-24. Since you couldn't exclude NO2-based reactions as major route of sulfate formation, the analysis of the relationship between SOR and NO2 is not necessary. (12) Page 9, line 2-3. The authors described on page 7, line 7-10 that the self-catalytic nature is beyond the scope of your study. However, you illustrate the importance of the self –catalytic in this paragraph. I think it's self-contradictory. (13) Page 10, line 21. Should be Zhejiang University.

---

## Referee Comment (RC2) · Anonymous Referee #1 · 9 May 2019

The manuscript by Fang et al. provides a nice year-long dataset of PM2.5 along with chemical composition and some important precursors, which would be of interest in improving the understanding of pollution evolution in Beijing. Throughout the manuscript, the authors focused mostly on the observed relationships between SOR and O3/RH, and made conclusions that O3 and RH are two "prerequisites" of sulfate formation. These conclusions, however, are predictable. RH and O3 together provide almost all the necessary conditions for sulfate formation: for gas phase oxidation, they are sources of OH, and for aqueous phase or heterogenous phase oxidations, water and oxidants (O3, H2O2 (O3 was a precursor of H2O2). This is saying, that the authors focused on the relationship between SOR and O3/RH and concluded on multi-phase reaction by H2O2 oxidation dominate (or major) sulfate formation is over concluded, or

more like a speculation, especially given the absence of H2O2 data.

In addition, there should be seasonal difference on the formation route, for example, in summer, pollution was the lowest and SOR was the highest, given the data presented, one cannot judge that multi-phase reaction by H2O2 oxidation should be responsible for sulfate formation: won't the gas-phase oxidation also enhanced in summer? In fact, for multiphase reactions, AWC might be a better indicator, however, as shown in Figure 7, SOR is not well correlated with AWC but better with RH. This for me is a good if not strong indicator that gas-phase oxidation (promoted by high O3 + RH + insulation) is important for at least summer high SOR.

Other Suggestions: 1) The fact of no correlation between SOR and NO2 could make a good argument on the role of NO2 in sulfate formation, I suggest to emphases this point. In addition, comparing SOR, NO2 and NH4+ (it would be better if NH3 is available), and see if there is any clue on the role of NH3 in aerosol pH and the promoted NO2 oxidation route as proposed by earlier studies.

2) It looks the authors dealt with SOR as a sole local phenomenon (local emission and local oxidation), but how about the difference in the regional transport of SO2 and SO42-? What would this do to SOR?

3) There is observational data on the relationship of H2O2 concentration and temperature in Beijing (Fu, A.: Study on peroxides concentration and its influencing factors in the urban atmosphere, master of engineering, College of Environmental and Resource Sciences, Zhejiang University, Hangzhou, China, 56 pp., 2014 (in Chinese) ), the authors can derive the H2O2 concentration from the temperature data to better constrain the role of H2O2 by comparisons with O3 and SOR data.

4) Atmospheric oxidation capacity is a rather vague (or big) definition when related to specific oxidation route of chemicals. Try to avoid

5) The manuscript need a little bit more tuned, e.g., line 31-32: what is "a given RH

threshold"?

---

## Referee Comment (RC3) · Anonymous Referee #3 · 12 May 2019

General points: This study provides long-term continuous filter sampling and composition analysis data of PM2.5. Many previous studies usually conducted such kind of observation intermittent for a short period, but such long-term uninterrupted observations are quite scarce. Thus, the data is of scientific value for analysis of variation characteristics of PM2.5 compositions and model validation. Moreover, this paper focus on identifying the possible factors on sulfate formation, which is helpful for understanding of mechanism of sulfate formation. If the general and specific points below are addressed, I recommend this paper for publication. 1. The authors investigate the relationship of SOR and RH/O3, and conclude that RH and O3 are two "prerequisite" for sulfate formation. But the further speculation of "H2O2 oxidation was proposed to be the major route" seems lack of sufficient evidence without the H2O2 data and

laboratory experiment results support. The refs. (Sievering et al. 2004; Alexander et al., 2005) are also not solid enough to back your speculation. 2. The authors should adjust the structures of the paper to make more clear and concise statement. Although the overview of the data is needed for the readers, the discussion in Sect3.1 is concentrated on the source appointment of PM2.5, which is abundant and deviate away from the theme. I suggest this Sect. discuss the variations of the components concentrations and contribution ratios using the classification method based on season or pollution levels. Sulfate can be focused on. 3. The order of the figures and tables in the main text and SI is confusing, the authors should rearrange the figures and tables according to the main text. 4. The authors should carefully go through the whole manuscript to avoid mistakes.

Specific points: 1. Avoid duplicated sentences and definitions. E.g. Page1, line18-20 vs Page 2, line 1-2; Page 1, line 25-26 vs Page2, line 23-26, and the definition of "self-catalytic" is vague. 2. Page 2, line 14, what is "various parameters" refer to? 3. Page 4, line 6, Figure 1 should be "Fig. 1"; Page 4, line 15, give the location information (lat, long) of the site; Page 5, line 4-10, rewrite the first sentence "The chemical . . . . . . (TEOs)." There actually 8 categories including "others" and the category is not according to the source type. Why you start with Fig. S2 not S1? Page 6 why you put Fig. 4 before Fig.3 in your text. Check the orders as mentioned in general points 3. 4. Sect. 3.2 How do you give the definition of threshold? The SOR or ΔSOR exceed certain value? The authors also compared the results with previous studies in this Sect., what is the reason for the difference in these studies? 5. Page 9, line 5-8 and Page 9, line 12-14 the sentences are contradictory. 6. Use "clear", "formation", "evolution" etc. to represent different pollution level is improper, because you do not conduct case or course study in the paper. 7. How about other factors such as wind speed and wind direction impact on SOR except RH and O3? 8. Is all the data in this paper daily data? Please give make it clear in the paper. 9. SOR is the conversion ratio of SO2, I doubt whether it can indicate the conversion rate (or speed) as you mentioned in your paper (e.g. Page 1, line 21, Page 10, line 14 etc.) What is the relationship of O3 and

atmospheric oxidative capacity? AWC and RH? Please reconsider in your statement and discussions? (Page 8, line 10, Page 9, line 10-11 etc.). 10. The fitting methods were used in this paper (Fig. 5 and Fig. S5), please give the evaluation parameters (such as p-value and R) of the fitting method to prove the validity and accuracy of the fitting. Also in Fig 5b, the last 2 box bins only have 1-2 points, does the results make sense? 11. Give the right form of the author's name in Page 1 and Page 12. There should be a space between units and the quantity.
* * *

---

## Author Comment (AC1) · 23 Jul 2019

Please see attached in the supplement the response to referee #1

———————————————

---

## Author Comment (AC2) · 23 Jul 2019

**Response to the Comments of Referees**

**RH and O₃ concentration as two prerequisites for sulfate formation**

Yanhua Fang and Chunxiang Ye, Junxia Wang, Yusheng Wu, Min Hu, Weili Lin, Fanfan Xu, Tong Zhu

We thank the referees for the critical comments, which are very helpful in improving the quality of the manuscript. We have made major revision based on the critical comments and suggestions of the referees. Our point-by-point responses to the comments are listed in the following.

**Anonymous Referee #2**

**Comment NO.1:** *The paper deals with the mass concentration and chemical composition of PM$_{2.5}$ in Beijing during 1 year from filter samples and its correlation with pollution classes (clear days, slight, light, medium and heavy pollution). Most of the paper is devoted to the two prerequisites for sulfate formation based discussion. This is certainly a positive feature of the paper. Although the article has a clear logical structure, I strongly recommend to make the text more concise, to clarify statements, and to delete redundancies.*

**Response:** Accepted.

We deleted redundancies and clarified several statements based on the referee's suggestions to make the text more concisely.

**Changes in Manuscript:** We have deleted redundancies in abstract and section 3.1, please refer to the revised manuscript, Page 1 lines 14–28 and Page 5 lines 8–15. We have replaced the atmospheric oxidation capacity to appropriate oxidant, please refer to the revised manuscript, Page 6 line 16, Page 8 line 31, Page 9 lines 23–24, and Page 10 line 8. Please also refer to the comments NO.5, NO.9, and NO.12.

**Comment NO.2:** *Most importantly, in the absence of data on hydrogen peroxide, all*

*speculation seems weak. The main idea of the article is still in the cognition of previous studies, and no more innovative conclusions have been put forward. In a word, this article is full of paradoxical conclusions and cannot provide a powerful help to the scientific community. Therefore, I don't recommend the publication in ACP journal in current status.*

**Response:** We have made major revision of our manuscript, concerning the following two point:

1) We would like to first summary the main contribution of our manuscript here. Our manuscript is the first to introduce the idea that there are some threshold values (or turning points), above which the SOR increases rapidly, for both RH and $O_3$, based on year-long observations. We presented clear observational evidence for these thresholds, best seen in the plot of SOR versus RH and $O_3$ data (Fig. 5 in the revised manuscript, Page 20). The thresholds at roughly 35 ppb $O_3$ and 45% RH are observed. Although such turning point possible varies in different seasons and locations, such thresholds immediately indicate that both RH and $O_3$ are two "prerequisites" for the multiphase formation of sulfate. In the case of the RH threshold, this is consistent with current understanding in the dependence of the multiphase sulfate formation on aerosol water, since RH threshold relates to the semisolid-to-liquid phase transition of atmospheric aerosols. Correlation analysis between SOR and AWC further backs this point up (Fig. R1 in this response, which has been added to the revised SI as Fig. S3, Page 6). In the case of $O_3$ concentration threshold, this is consistent with the consumption of liquid oxidants in multiphase sulfate formation.

[Figure]

**Figure R1**. Plot of the sulfur oxidation ratio (SOR) against aerosol water content (AWC) (note log scale), grouped by $O_3$ concentration. The solid blue circles represent $O_3 > 35$ ppb and the solid black circles represent $O_3 < 35$ ppb. The boxes represent, from top to bottom, the $75^{th}$, $50^{th}$, and $25^{th}$ percentiles in each bin, which were also separated according to the 35 ppb $O_3$ concentration threshold; the bin widths were set such that there were an approximately equal number of data points in each bin. The whiskers, solid squares, and open circles represent 1.5 times the interquartile range (IQR), mean values, and outlier data points, respectively. The lines are best fits to the mean values based on a sigmoid function. Data for days with rain or snow were excluded from this plot.

2) We agree with the referee that lack of $H_2O_2$ measurement is a weakness in the discussion of possible role of $H_2O_2$ in sulfate formation mechanisms. To add more confidence in such discussion, a proxy measurement of $H_2O_2$ is included in the revised manuscript. Taking the advice of referee #1, that $H_2O_2$ was non-linearly correlated with temperature (Fu, 2014). $H_2O_2$ was estimated from temperature, by assuming the same relationship applicable to our measurements in the full year of 2012–2013. As shown in Fig.S2 in this response (added in the revised SI as Fig. S6, Page 9), maximum concentration of $H_2O_2$ in summer is expected and confirmed, which is in line with the fastest sulfate formation in summer all over the year. SOR was further plotted against $H_2O_2$ and positive correlation was found between them (Fig. R3 in this response, which has been added in the revised SI as Fig.S7, Page 9.). In addition, coincident increases in the concentration of $H_2O_2$ and $PM_{2.5}$ in winter of Beijing also lead to an important role of the $H_2O_2$ route in sulfate

formation (Ye et al., 2018). These discussions were added up to our previous analysis in the original manuscript, i.e., $O_3$ and $H_2O_2$ are proposed to be the major oxidants in multiphase sulfate formation based on the above threshold analysis. Since $O_3$ was excluded as a major oxidant in multiphase sulfate formation, for that the high aerosol acidity in urban environments limits its reaction rate, $H_2O_2$ remains the only possible liquid phase oxidant (Page 7 lines 14–24 in the revised manuscript). Based on all the above discussions, we carefully proposed in the revised manuscript that $H_2O_2$ might be an important oxidant of sulfate formation.

[Figure]

**Figure R2**. Time series of estimated $H_2O_2$ from from March 12012 to February 28 2013 (open black circles). $H_2O_2$ was estimated from temperature (T) based on the fitting function $H_2O_2 = 0.1155e^{0.0846T}$ according to Fu (2014). The boxes represent, from top to bottom, the 75th, 50th, and 25th percentiles for each season. The whiskers, solid red squares, and open red circles represent 1.5 times the interquartile range (IQR), seasonal mean values, and outlier data points, respectively.

[Figure]

**Figure R3**. Plot of the SOR against estimated $H_2O_2$ grouped by RH. The solid blue circles represent RH > 45 % and the solid black circles represent RH < 45 %. The boxes represent, from top to bottom, the 75th, 50th, and 25th percentiles in each bin. The bin widths were set such that there were an approximately equal number of data points in each bin. The whiskers, solid squares, and open circles represent 1.5 times the

IQR, mean values, and outlier data points, respectively. The line are best fits to the mean values based on an exponential function. Data for days with rain were excluded from this plot.

**Changes in Manuscript:** A summary of our scientific contribution has been revised in the abstract and in the text, please refer to the revised manuscript, Page 1 lines 13–19 and Page 5 lines 25–26. Further discussions on the role of $H_2O_2$ has also been added to the revised manuscript, Page 7 lines 14–24.

**Comment NO.3:** *The author name should be Weili Lin.*

**Response:** Accepted.

**Changes in Manuscript:** We have made a correction, please refer to the revised manuscript, Page 1 line 2.

**Comment NO.4:** *"threshold of RH and ozone" Where is this statement coming from? Is it a definition/estimate of the authors? If the threshold changed with different locations and seasons? What is the effect of these thresholds?*

**Response:**

1) "Thresholds of RH and ozone" are obtained based our measurement in the full year of 2012-2013 that above some turning points of RH and $O_3$ concentration, SORs increase rapidly. This is best seen in the plot of SOR versus RH and $O_3$ data (Fig. 5 in the original manuscript, Page 20). Our interpretation of this is that there are thresholds or turning points in RH and $O_3$ concentration that must be exceeded to allow for the fast formation of sulfate. Although such turning point possible varies in different seasons and locations, such thresholds immediately indicate that both RH and $O_3$ are two "prerequisites" for the multiphase formation of sulfate.

2) It is also the authors' interpretation that the threshold of RH is around 45 % and the threshold of $O_3$ is around 35 ppb. There could be some uncertainty attached with such inferred values. For example, one could argue that the threshold of $O_3$ concentration is any value between 30–40 ppb. Also, the daily average RH and $O_3$ data used in our analyses are not the best to evaluate the thresholds. For example, the observed RH threshold is proposed to be determined by the phase transition RH.

However, the timescale of the phase transition in ambient air is on the order of seconds (Liu et al., 2008), in comparison to RH changes on timescales of hours to days, and thus the daily average RH is not an accurate estimate of the phase transition RH. This explains why the apparent RH threshold of 45 % observed in Fig. 5 is somewhat below the *in situ* phase transition RH of 50–60 % (Liu et al., 2017b).

3) The thresholds might change with locations and seasons. For instance, Fig. R4 in this response (added to the revised manuscript as Fig. 6, Page 21) suggests that the RH threshold is roughly around 45 % during all four seasons in Beijing. The turning point varied within 40%- 50% in different sampling location of Beijing (Liu et al., 2015; Xu et al., 2017; Yang et al., 2015; Zheng et al., 2015). However, similar analyses must be performed using high time resolution data to confirm the trends observed based on our daily average data.

[Figure]

**Figure R4**. Plots of SORs against RH, grouped by $O_3$ concentration in four seasons. The solid blue circles represent $O_3 > 35$ ppb and the solid black circles represent $O_3 < 35$ ppb. The boxes represent, from top to bottom, the 75th, 50th, and 25th percentiles in each bin ($\Delta RH = 5$ %). The whiskers, solid red squares, and open red circles represent 1.5 times the IQR, mean values, and outlier data points, respectively. The red lines are best fits to mean values based on sigmoid function. Data for days with rain or snow were excluded from these plots.

4) As stated above, above the thresholds of RH and $O_3$ concentration, sulfate formation could be enhanced (Please also refer to the response of comment NO.2).

**Changes in the Manuscript:** A discussion on the possible seasonal variations in the thresholds were added in our revised manuscript, please refer to the revised manuscript, Page 6 lines 32–34 and Page 7 lines 1–7.

**Comment NO.5:** *Redundancy: Page 1 line 15-16 and line 24-25. Line 13-14 and Line 17-18.*

**Response:** Accepted.

**Changes in the Manuscript:** We have rewritten the abstract and deleted the redundant sentences in the revised manuscript. Please refer to the revised manuscript, Page 1 lines 14–28.

**Comment NO.6:** *Section 2.1.2. Please add the steps of weighing after sampling.*

**Response:**
The steps of weighting after sampling have been provided in the original manuscript. Please refer to the revised manuscript, Page 4 lines 3–5 (highlighted).

**Comment NO.7:** *Page 4, line 27. Should be annual standard*

**Response:** Accepted.

**Changes in Manuscript:** We have changed the phrase to "Chinese National Ambient Air Standard annual mean concentration of ", please refer to the revised manuscript, Page 5 line 5.

**Comment NO.8:** *Page 5, line 2. The method to calculate POM should be introduced in previous section.*

**Response:**
The method to calculate POM was provided in the original SI. The discussion on source appointment, including POM, has been deleted in the revised manuscript and SI.

**Comment NO.9:** *Overall, section 3.1 is not necessary, because it has nothing to do*

*with the main idea. If this section is deleted in the main article, it will not affect the presentation of the article. For example, the authors described the measurements of ions, organics and metal. However, ions except SNA, organics and metals except Fe didn't help the discussion of your topic. Therefore, the method and results section should to be streamlined.*

**Response:** Accepted

**Changes in Manuscript:** Sect 3.1 has been reduced so that a general description of data is presented, and that variations in $PM_{2.5}$ and its main components are introduced. Please refer to the revised manuscript, Page 5 lines 3–18.

**Comment NO.10:** *Section 3.2. I strongly recommend the authors discussing the relationship between sulfate and RH/ozone in different seasons. The threshold should be changed with seasons.*

**Response:** Accepted

**Changes in Manuscript:** The seasonal variations are discussed now in the revised manuscript (also refer to response to comment NO.4). Please refer to the revised manuscript, Page 6 lines 32–34 and Page 7 lines 1–7.

**Comment NO.11:** *Page 7, line 12-16 repeats the previous statement.*

**Response:**
We intended to summarise our major findings and discuss their implications in this section.

**Changes in Manuscript:** We have rewritten the sentences, please refer to the revised manuscript, Page 7 lines 12–24.

**Comment NO.12:** *Page 7, line 14. What is the atmospheric oxidative capacity? From your statement, does ozone concentration correspond to this? Is it correct? Do you have some references to support your opinion? The authors should clarify this question because the same definition is also used in Page 9, line 20.*

**Response:** Accepted.

Atmospheric oxidative capacity relates to the concentrations of major oxidants such as OH radicals, $O_3$, etc. (Murray et al., 2009). Since $O_3$ is a major oxidant and a precursor to other major oxidants, including OH radicals, to a certain degree, $O_3$ can be used as a proxy for atmospheric oxidative capacity. To improve clarity, atmospheric oxidative capacity was replaced by the appropriate oxidant in each context in the revised manuscript.

**Changes in Manuscript:** Atmospheric oxidative capacity was replaced by the appropriate oxidant. Please refer to the revised manuscript, Page 6 line 16, Page 8 line 31, Page 9 lines 23–24, and Page 10 line 8.

**Comment NO.13:** *Page 7, Line 23-24. Since you couldn't exclude $NO_2$-based reactions as major route of sulfate formation, the analysis of the relationship between SOR and $NO_2$ is not necessary.*

**Response:**

We took the advice of referee #1 and further discussed the possible role of $NO_2+O_2$ route in the revised manuscript based on two points. First, no correlation between the SOR and $NO_2$ was found. Secondly, although in our study, $NH_3$ measurements were not available, previous studies has reported a mean aerosol pH value of ~4.2 with a low limit of ~3.0 in Beijing (Ding et al., 2019; Liu et al., 2017a), which suggests that several routes of sulfate formation, such as $NO_2 + O_2$, TMIs $+ O_2$, $O_3$ etc., are suppressed. Therefore, we proposed that $NO_2+O_2$ might not be a major mechanism of sulfate formation.

**Changes in Manuscript:** Please refer to the revised manuscript, Page 7 lines 30–32 and Page 8 lines 1–3.

**Comment NO.14:** *Page 9, line 2-3. The authors described on page 7, line 7-10 that the self-catalytic nature is beyond the scope of your study. However, you illustrate the importance of the self–catalytic in this paragraph. I think it's self-contradictory.*

**Response:**

To clarify, our manuscript states that the self-constrained nature, i.e., sulfate formation increases the acidity of aerosols, which suppresses sulfate formation via several routes, such the $O_3$ oxidation and TMIs + $O_2$ routes. The self-catalytic nature of sulfate formation is best seen from the perspective that sulfate formation adds up the aerosol volume/surface density which helps with further sulfate formation. Those two mechanisms compete in determining the sulfate formation as pollution accumulation. In our manuscript, the self-constrained nature of sulfate formation is not discussed in detail due to the lack of direct or proxy measurements of aerosol acidity in our measurements.

**Comment NO.15:** *Page 10, line 21. Should be Zhejiang University.*

**Response:** Accepted.

**Changes in Manuscript:** We have made the correction. Please refer to the revised manuscript, Page 11 lines 18–19.

**References**

Ding, J., Zhao, P., Su, J., Dong, Q., Du, X., and Zhang, Y.: Aerosol pH and its driving factors in Beijing, Atmos. Chem. Phys., 19, 7939-7954, https://doi.org/10.5194/acp-19-7939-2019, 2019.

Fu, A.: Study on peroxide concentration and its influence factors in the urban atmosphere, Master, College of Environmental and Resource Sciences, Zhejiang University, Hangzhou, China, 2014 (in Chinese).

Liu, M., Song, Y., Zhou, T., Xu, Z., Yan, C., Zheng, M., Wu, Z., Hu, M., Wu, Y., and Zhu, T.: Fine particle pH during severe haze episodes in northern China, Geophys. Res. Lett., 44, 5213-5221, https://doi.org/10.1002/2017GL073210, 2017a.

Liu, X., Sun, K., Qu, Y., Hu, M., Sun, Y., Zhang, F., and Zhang, Y.: Secondary formation of sulfate and nitrate during a haze episode in megacity Beijing, China, Aerosol Air Qual. Res., 2246 - 2257, https://doi.org/10.4209/aaqr.2014.12.0321, 2015.

Liu, Y. C., Wu, Z. J., Wang, Y., Xiao, Y., Gu, F. T., Zheng, J., Tan, T. Y., Shang, D. J., Wu, Y. S., Zeng, L. M., Hu, M., Bateman, A. P., and Martin, S. T.: Submicrometer particles are in the liquid state during heavy haze episodes in the urban atmosphere of Beijing, China, Environ. Sci. Technol. Lett., 4, 427-432, https://doi.org/10.1021/acs.estlett.7b00352, 2017b.

Liu, Y. J., Zhu, T., Zhao, D. F., and Zhang, Z. F.: Investigation of the hygroscopic properties of $Ca(NO_3)_2$ and internally mixed $Ca(NO_3)_2/CaCO_3$ particles by micro-

Raman spectrometry, Atmos. Chem. Phys., 8, 7205-7215, https://doi.org/10.5194/acp-8-7205-2008, 2008.

Murray, L. T., Mickley, L., Kaplan, J. O., Sofen, E. D., Alexander, B., Jones, D. B., and Jacob, D. J.: Evolution of the oxidative capacity of the troposphere since the Last Glacial Maximum, 3589-3622, https://doi.org/10.5194/acp-14-3589-2014, 2009.

Xu, L., Duan, F., He, K., Ma, Y., Zhu, L., Zheng, Y., Huang, T., Kimoto, T., Ma, T., Li, H., Ye, S., Yang, S., Sun, Z., and Xu, B.: Characteristics of the secondary water-soluble ions in a typical autumn haze in Beijing, Environ. Pollut., 227, 296-305, https://doi.org/10.1016/j.envpol.2017.04.076, 2017.

Yang, Y. R., Liu, X. G., Qu, Y., An, J. L., Jiang, R., Zhang, Y. H., Sun, Y. L., Wu, Z. J., Zhang, F., Xu, W. Q., and Ma, Q. X.: Characteristics and formation mechanism of continuous hazes in China: a case study during the autumn of 2014 in the North China Plain, Atmos. Chem. Phys., 15, 8165-8178, https://doi.org/10.5194/acp-15-8165-2015, 2015.

Ye, C., Liu, P., Ma, Z., Xue, C., Zhang, C., Zhang, Y., Liu, J., Liu, C., Sun, X., and Mu, Y.: High $H_2O_2$ concentrations observed during haze periods in wintertime of Beijing: Importance of $H_2O_2$-oxidation in sulfate formation, Environ. Sci. Technol. Lett., https://doi.org/10.1021/acs.estlett.8b00579, 2018.

Zheng, G. J., Duan, F. K., Su, H., Ma, Y. L., Cheng, Y., Zheng, B., Zhang, Q., Huang, T., Kimoto, T., Chang, D., Poschl, U., Cheng, Y. F., and He, K. B.: Exploring the severe winter haze in Beijing: the impact of synoptic weather, regional transport and heterogeneous reactions, Atmos. Chem. Phys., 15, 2969-2983, https://doi.org/10.5194/acp-15-2969-2015, 2015.

---

## Author Comment (AC3) · 23 Jul 2019

**Response to the Comments of Referees**

**RH and O₃ concentration as two prerequisites for sulfate formation**

Yanhua Fang and Chunxiang Ye, Junxia Wang, Yusheng Wu, Min Hu, Weili Lin, Fanfan Xu, Tong Zhu

We thank the referees for the critical comments, which are very helpful in improving the quality of the manuscript. We have made major revision based on the critical comments and suggestions of the referees. Our point-by-point responses to the comments are listed in the following.

**Anonymous Referee #3**

**Comment NO.1:** *General points: This study provides long-term continuous filter sampling and composition analysis data of PM$_{2.5}$. Many previous studies usually conducted such kind of observation intermittent for a short period, but such long-term uninterrupted observations are quite scarce. Thus, the data is of scientific value for analysis of variation characteristics of PM$_{2.5}$ compositions and model validation. Moreover, this paper focus on identifying the possible factors on sulfate formation, which is helpful for understanding of mechanism of sulfate formation. If the general and specific points below are addressed, I recommend this paper for publication.*

*The authors investigate the relationship of SOR and RH/O₃, and conclude that RH and O₃ are two "prerequisite" for sulfate formation. But the further speculation of "H₂O₂ oxidation was proposed to be the major route" seems lack of sufficient evidence without the H₂O₂ data and laboratory experiment results support. The refs. (Sievering et al. 2004; Alexander et al., 2005) are also not solid enough to back your speculation.*

**Response:** We are grateful to the reviewer for the positive and encouraging comments on the dataset and the scientific contribution of our manuscript to understanding sulfate

formation.

We agree with the referee that lack of $H_2O_2$ measurement is a weakness in the discussion of possible role of $H_2O_2$ in sulfate formation mechanisms. To add more confidence in such discussion, a proxy measurement of $H_2O_2$ is included in the revised manuscript. Taking the advice of referee #1, that $H_2O_2$ was non-linearly correlated with temperature (Fu, 2014). $H_2O_2$ was estimated from temperature, by assuming the same relationship applicable to our measurements in the full year of 2012–2013. As shown in Fig.S1 in this response (added in the revised SI as Fig. S6, Page 9), maximum concentration of $H_2O_2$ in summer is expected and confirmed, which is in line with the fastest sulfate formation in summer all over the year. SOR was further plotted against $H_2O_2$ and positive correlation was found between them (Fig. R2 in this response, which has been added in the revised SI as Fig.S7, Page 9.). In addition, coincident increases in the concentration of $H_2O_2$ and $PM_{2.5}$ in winter of Beijing also lead to an important role of the $H_2O_2$ route in sulfate formation (Ye et al., 2018). These discussions were added up to our previous analysis in the original manuscript, i.e., $O_3$ and $H_2O_2$ are proposed to be the major oxidants in multiphase sulfate formation based on the above threshold analysis. Since $O_3$ was excluded as a major oxidant in multiphase sulfate formation, for that the high aerosol acidity in urban environments limits its reaction rate, $H_2O_2$ remains the only possible liquid phase oxidant (Page 7 lines 14–24 in the revised manuscript). Based on all the above discussions, we carefully proposed in the revised manuscript that $H_2O_2$ might be an important oxidant of sulfate formation.

[Figure]

**Figure R1**. Time series of estimated $H_2O_2$ from from March 12012 to February 28 2013 (open black circles). $H_2O_2$ was estimated from temperature (T) based on the fitting function $H_2O_2 = 0.1155e^{0.0846T}$ according to Fu (2014). The boxes represent, from top to bottom, the 75th, 50th, and 25th percentiles for

each season. The whiskers, solid red squares, and open red circles represent 1.5 times the interquartile range (IQR), seasonal mean values, and outlier data points, respectively.

[Figure]

**Figure R2**. Plot of the SOR against estimated $H_2O_2$ grouped by RH. The solid blue circles represent RH > 45 % and the solid black circles represent RH < 45 %. The boxes represent, from top to bottom, the 75th, 50th, and 25th percentiles in each bin. The bin widths were set such that there were an approximately equal number of data points in each bin. The whiskers, solid squares, and open circles represent 1.5 times the IQR, mean values, and outlier data points, respectively. The line are best fits to the mean values based on an exponential function. Data for days with rain were excluded from this plot.

**Changes in Manuscript:** Discussions on the role of $H_2O_2$ has also been added to the revised manuscript, Page 7 lines 14–24.

**Comment NO.2:** *The authors should adjust the structures of the paper to make more clear and concise statement. Although the overview of the data is needed for the readers, the discussion in Sect3.1 is concentrated on the source appointment of PM$_{2.5}$, which is abundant and deviate away from the theme. I suggest this Sect. discuss the variations of the components concentrations and contribution ratios using the classification method based on season or pollution levels. Sulfate can be focused on.*

**Response:** Accepted

**Changes in Manuscript:** Sect 3.1 has been reduced so that a general description of data is presented, and that variations in PM$_{2.5}$ and its main components are introduced. Please refer to the revised manuscript, Page 5 lines 3–18.

**Comment NO.3:** *The order of the figures and tables in the main text and SI is confusing, the authors should rearrange the figures and tables according to the main text.*

**Response:** Accepted

**Changes in Manuscript:** We have rearranged the figures. Please refer to the revised manuscript, Pages 18–19 Figs 3–4.

**Comment NO.4:** *The authors should carefully go through the whole manuscript to avoid mistakes. Specific points: 1. Avoid duplicated sentences and definitions. E.g. Page1, line18- 20 vs Page 2, line 1-2; Page 1, line 25-26 vs Page2, line 23-26, and the definition of "self-catalytic" is vague.*

**Response:** Accepted

1) Duplicated sentences deleted in the revised manuscript.
2) We need to better define the term "self-catalytic" as referee #2 has also suggested. We have therefore defined it consistently in both the abstract and introduction. The definition has changed to: "the formation of hydrophilic sulfate aerosols under high RH conditions results in an increase in aerosol water content, which results in greater particle volume for further multiphase sulfate formation".

**Changes in Manuscript:** The definition has been clarified, please refer to the revised manuscript, Page 1 lines 25–27 and Page 2 lines 16–18.

**Comment NO.5:** *Page 2, line 14, what is "various parameters" refer to*

**Response:** oxidants, catalysts, meteorological conditions, etc.

**Changes in Manuscript:** We have clarified the parameters as "exactly how do various parameters (oxidants, catalysts, meteorological conditions, etc.) influence sulfate formation" in the revised manuscript, Page 2 line 10.

**Comment NO.6:** *Page 4, line 6, Figure 1 should be "Fig. 1"; Page 4, line 15, give the location information (lat, long) of the site; Page 5, line 4-10, rewrite the first sentence "The chemical. . . . . . (TEOs)." There actually 8 categories including "others" and the category is not according to the source type. Why you start with Fig. S2 not S1? Page 6 why you put Fig. 4 before Fig.3 in your text. Check the orders as mentioned in general points 3.*

**Response:** Accepted.

**Changes in Manuscript:**

1) Figure 1 has been changed to Fig. 1. Please refer to the revised manuscript, Page 3 line 22.

2) The lat/long of the Beijing Meteorological Observatory Station (116.47° E, 39.81° N) has been added. Please refer to the revised manuscript, Page 3 line 29.

3) The sentence the reviewer mentions has been rewritten to: "The chemical components of $PM_{2.5}$ were divided into eight categories: sulfate, nitrate, ammonium, organic matter (OM), EC, minerals, trace element oxides (TEOs), and others." Please refer to the revised manuscript, Page 4 lines 14–15.

4) We have rearranged the order of Figs. Please refer to the revised manuscript, Page 4 lines 17–18 and Pages 18–19 Figs. 3–4.

**Comment NO.7:** *Sect. 3.2 How do you give the definition of threshold? The SOR or ΔSOR exceed certain value? The authors also compared the results with previous studies in this Sect., what is the reason for the difference in these studies?*

**Response:**

1) "Thresholds of RH and ozone" are obtained based our measurement in the full year of 2012-2013 that above some turning points of RH and $O_3$ concentration, SORs increase rapidly. This is best seen in the plot of SOR versus RH and $O_3$ data (Fig. 5 in the original manuscript, Page 20). Our interpretation of this is that there are thresholds or turning points in RH and $O_3$ concentration that must be exceeded to allow for the fast formation of sulfate. Although such turning point possible varies in different seasons and locations, such thresholds immediately indicate that both RH and $O_3$ are two "prerequisites" for the multiphase formation of sulfate.

2) It is also the authors' interpretation that the threshold of RH is around 45 % and the threshold of $O_3$ is around 35 ppb. There could be some uncertainty attached with such inferred values. For example, the thresholds might change with locations and seasons. Also, the daily average RH and $O_3$ data used in our analyses are not the best to evaluate the thresholds. For example, the observed RH threshold is proposed

to be determined by the phase transition RH. However, the timescale of the phase transition in ambient air is on the order of seconds (Liu et al., 2008), in comparison to RH changes on timescales of hours to days, and thus the daily average RH is not an accurate estimate of the phase transition RH. This explains why the apparent RH threshold of 45 % observed in Fig. 5 is somewhat below the *in situ* phase transition RH of 50–60 % (Liu et al., 2017).

**Comment NO.8:** *Page 9, line 5-8 and Page 9, line 12-14 the sentences are contradictory*

**Response:**

The sentences on Page 9, lines 5–8 explain that the self-catalytic nature of sulfate formation accounts for the increased SOR as pollution accumulates. The sentences on page 9, lines 12–14 summarise our conclusion about the thresholds of $O_3$ and RH.

**Comment NO.9:** *Use "clear", "formation", "evolution" etc. to represent different pollution level is improper, because you do not conduct case or course study in the paper.*

**Response:** Accepted.

**Changes in Manuscript:** The definitions have been changed to: clean, moderate pollution, heavy pollution, and severe pollution in the revised manuscript. Please refer to the revised manuscript, Page 9 line4, Page 25 Fig. 10, and Page 26 Fig.11. These still represent each quartile of $PM_{2.5}$ levels.

**Comment NO.10:** *How about other factors such as wind speed and wind direction impact on SOR except RH and $O_3$?*

**Response:**

Wind speed and wind direction are not assumed to be influencing parameters of sulfate formation according to the mechanism summarised in the introduction section and hence were not discussed in our manuscript. However, it is clear that high SORs and high $PM_{2.5}$ were commonly found at low to medium wind speeds (Fig. R3 in this

response), which might be related to the increasing SORs as aerosol pollution accumulated. Hotspots of SOR at high wind speed with northwest sector and south sector are also found, which might be related to regional transport of sulfate. The uncertainty concerning regional transport has been discussed in the response to referee #1 comment NO.3.

[Figure]

**Figure R3**. Bivariate polar plots for (a) SOR and (b) PM$_{2.5}$. The grey shading indicates lack of data. Wind speed and wind direction were download from the National Climate Data Center (www.ncdc.noaa.gov), which were measured at a station located in the Beijing Capital International Airport.

**Comment NO.11:** *Is all the data in this paper daily data? Please give make it clear in the paper.*

**Response:**

Yes, all the data used in this manuscript are daily averages and this has been clarified in the method section of the revised manuscript (Page 3, lines 29–30 in the revised manuscript).

To be more specific, daily PM$_{2.5}$ filter samples were collected for 23.5 h, from 9:30 am to 9:00 am the next day; thus, PM$_{2.5}$ and its components were daily averaged data. Gaseous pollutants (SO$_2$, O$_3$, NO$_x$, etc.) and RH with a time resolution of mins were averaged according to the filter sampling time period. Daily solar radiation data was used as it is.

**Comment NO.12:** *SOR is the conversion ratio of SO$_2$, I doubt whether it can indicate the conversion rate (or speed) as you mentioned in your paper (e.g. Page 1, line 21,*

*Page 10, line 14 etc.)    What is the relationship of O₃ and atmospheric oxidative capacity? AWC and RH? Please reconsider in your statement and discussions? (Page 8, line 10, Page 9, line 10-11 etc.).*

**Response:**

1) We agree with the referee that SOR is defined as the ratio of sulfate to total sulfur and it is not the $SO_2$-sulfate conversion rate. However, due to the long chemical lifetime of sulfate, sulfate is tend to accumulate with chemical production within at least 24 hrs, which could be best reflected in SOR, the ratio of sulfate to total sulfur. SOR has been widely used as an indicator of $SO_2$-to-sulfate conversion in numbers of references (Sun et al., 2014; Zheng et al., 2015), where a high SOR reflects a high $SO_2$-to-sulfate conversion rate on average during the measurement period.

2) Atmospheric oxidative capacity relates to the concentrations of major oxidants such as OH radicals, $O_3$, etc. (Murray et al., 2009). Since $O_3$ is a major oxidant and a precursor to other major oxidants, including OH radicals, to a certain degree, $O_3$ can be used as a proxy for atmospheric oxidative capacity. To improve clarity, atmospheric oxidative capacity was replaced by the appropriate oxidant in each context in the revised manuscript.

3) The AWC calculated using the ISORROPIA-II thermodynamic model (http://isorropia.eas.gatech.edu). Please also refer to the revised SI (Page 3 lines 14-16). In brief, AWC is a function of aerosol mass concentration, aerosol chemical composition, RH, etc.

**Changes in Manuscript:** Atmospheric oxidative capacity was replaced by the appropriate oxidants. Please refer to the revised manuscript, Page 6 line 16, Page 8 line 31, Page 9 lines 23–24, and Page 10 line 8.

**Comment NO.13:** *The fitting methods were used in this paper (Fig. 5 and Fig. S5), please give the evaluation parameters (such as p-value and R) of the fitting method to prove the validity and accuracy of the fitting. Also in Fig 5b, the last 2 box bins only have 1-2 points, does the results make sense?*

**Response:** Accepted

1) $R^2$ has been added to Fig. 5 in the revised manuscript (Page 20).

2) In Fig. 5 (Page 20 in the revised manuscript), $O_3$ concentrations were grouped by 5 ppb intervals and RH by 5 % intervals. There were only a few data points on the right-hand sides of these figures because there were only a few days with daily average $O_3$ (RH) above 70 ppb (70 %). However, the shapes of the fits are not much different when we group them by the number of data points in each bin, as show in Fig. R4 in this response. $O_3$ in Fig. R4a was the original method that grouped by 5 ppb intervals, while $O_3$ in Fig. R4b were grouped with an approximately equal number of data points (15-16) in each bin, which shows the robustness of our fitting.

[Figure]

**Figure R4**. Plots of the SOR against $O_3$, grouped by RH. The solid blue circles represent RH > 45 % and the solid black circles represent RH < 45 %. The boxes represent, from top to bottom, the 75th, 50th, and 25th percentiles in each bin ((a) $\Delta O_3$ = 5 ppb, (b) variable $\Delta O_3$, 15–16 data points in each bin). The whiskers, solid red squares, and open red circles represent 1.5 times IQR, mean values, and outlier data points, respectively. The red lines are best fits to the mean values based on a sigmoid function. Data for days with rain or snow were excluded from these plots.

**Changes in Manuscript:** $R_2$ has been added to the plots that containing fitting lines. Please refer to the revised manuscript, Page 20 Fig. 5, Page 21 Fig. 6, Page 22 Fig. 7. Please also refer to the revised SI, Page 6 Fig. S3, and Page 9 Fig. S7.

**Comment NO.14:** *Give the right form of the author's name in Page 1 and Page 12. There should be a space between units and the quantity.*

**Response:** Accepted.

**Changes in Manuscript:**

1) The right form of the author's name has been given. Please refer to the revised manuscript, Page 1 line 2.

2) Space has been added between number and % or number between °C, Please refer to the revised manuscript, Page 1 line13, Page 4 lines 6 and 29, Page 5 lines 5, 14, 16, 23, 24, 26, and 29, Page 6 lines 28–29, Page 8 lines 21–22 and 29, Page 9 lines 3–4 and 32, Page 20 lines 4–5 and legend of Fig. 5b, Page 24 line 5, and Page 26 line 6.

**References**

Fu, A.: Study on peroxide concentration and its influence factors in the urban atmosphere, Master, College of Environmental and Resource Sciences, Zhejiang University, Hangzhou, China, 2014 (in Chinese).

Liu, Y. C., Wu, Z. J., Wang, Y., Xiao, Y., Gu, F. T., Zheng, J., Tan, T. Y., Shang, D. J., Wu, Y. S., Zeng, L. M., Hu, M., Bateman, A. P., and Martin, S. T.: Submicrometer particles are in the liquid state during heavy haze episodes in the urban atmosphere of Beijing, China, Environ. Sci. Technol. Lett., 4, 427-432, https://doi.org/10.1021/acs.estlett.7b00352, 2017.

Liu, Y. J., Zhu, T., Zhao, D. F., and Zhang, Z. F.: Investigation of the hygroscopic properties of $Ca(NO_3)_2$ and internally mixed $Ca(NO_3)_2/CaCO_3$ particles by micro-Raman spectrometry, Atmos. Chem. Phys., 8, 7205-7215, https://doi.org/10.5194/acp-8-7205-2008, 2008.

Murray, L. T., Mickley, L., Kaplan, J. O., Sofen, E. D., Alexander, B., Jones, D. B., and Jacob, D. J.: Evolution of the oxidative capacity of the troposphere since the Last Glacial Maximum, 3589-3622, https://doi.org/10.5194/acp-14-3589-2014, 2009.

Sun, Y. L., Jiang, Q., Wang, Z. F., Fu, P. Q., Li, J., Yang, T., and Yin, Y.: Investigation of the sources and evolution processes of severe haze pollution in Beijing in January 2013, J. Geophys. Res. Atmos., 119, 4380-4398, https://doi.org/10.1002/2014JD021641, 2014.

Ye, C., Liu, P., Ma, Z., Xue, C., Zhang, C., Zhang, Y., Liu, J., Liu, C., Sun, X., and Mu, Y.: High $H_2O_2$ concentrations observed during haze periods in wintertime of Beijing: Importance of $H_2O_2$-oxidation in sulfate formation, Environ. Sci. Technol. Lett., https://doi.org/10.1021/acs.estlett.8b00579, 2018.

Zheng, G. J., Duan, F. K., Su, H., Ma, Y. L., Cheng, Y., Zheng, B., Zhang, Q., Huang, T., Kimoto, T., Chang, D., Poschl, U., Cheng, Y. F., and He, K. B.: Exploring the severe winter haze in Beijing: the impact of synoptic weather, regional transport and heterogeneous reactions, Atmos. Chem. Phys., 15, 2969-2983, https://doi.org/10.5194/acp-15-2969-2015, 2015.

---

## Author Comment (AC4) · 23 Jul 2019

**Response to the Comments of Referees**

**RH and $O_3$ concentration as two prerequisites for sulfate formation**

Yanhua Fang and Chunxiang Ye, Junxia Wang, Yusheng Wu, Min Hu, Weili Lin, Fanfan Xu, Tong Zhu

We thank the referees for the critical comments, which are very helpful in improving the quality of the manuscript. We have made major revision based on the critical comments and suggestions of the referees. Our point-by-point responses to the comments are listed in the following.

**Anonymous Referee #1**

**Comment NO.1:** *The manuscript by Fang et al. provides a nice year-long dataset of $PM_{2.5}$ along with chemical composition and some important precursors, which would be of interest in improving the understanding of pollution evolution in Beijing. Throughout the manuscript, the authors focused mostly on the observed relationships between SOR and $O_3$/RH, and made conclusions that $O_3$ and RH are two "prerequisites" of sulfate formation. These conclusions, however, are predictable. RH and $O_3$ together provide almost all the necessary conditions for sulfate formation: for gas phase oxidation, they are sources of OH, and for aqueous phase or heterogeneous phase oxidations, water and oxidants ($O_3$, $H_2O_2$ ($O_3$ was a precursor of $H_2O_2$).*

*This is saying, that the authors focused on the relationship between SOR and $O_3$/RH and concluded on multi-phase reaction by $H_2O_2$ oxidation dominate (or major) sulfate formation is over concluded, or more like a speculation, especially given the absence of $H_2O_2$ data.*

*In addition, there should be seasonal difference on the formation route, for example, in summer, pollution was the lowest and SOR was the highest, given the data presented, one cannot judge that multi-phase reaction by $H_2O_2$ oxidation should be responsible for sulfate formation: won't the gas-phase oxidation also enhanced in summer? In fact,*

*for multiphase reactions, AWC might be a better indicator, however, as shown in Figure 7, SOR is not well correlated with AWC but better with RH. This for me is a good if not strong indicator that gas-phase oxidation (promoted by high O₃ + RH + insulation) is important for at least summer high SOR.*

**Response:**

We are grateful to the reviewer for the positive and encouraging comments on the dataset and the scientific contribution of our manuscript to understanding sulfate formation.

1) We would like to first summary the main contribution of our manuscript here. Our manuscript is the first to introduce the idea that there are some threshold values (or turning points), above which the SOR increases rapidly, for both RH and $O_3$, based on year-long observations. We presented clear observational evidence for these thresholds, best seen in the plot of SOR versus RH and $O_3$ data (Fig. 5 in the revised manuscript, Page 20). The thresholds at roughly 35 ppb $O_3$ and 45% RH are observed. Although such turning point possible varies in different seasons and locations, such thresholds immediately indicate that both RH and $O_3$ are two "prerequisites" for the multiphase formation of sulfate. In the case of the RH threshold, this is consistent with current understanding in the dependence of the multiphase sulfate formation on aerosol water, since RH threshold relates to the semisolid-to-liquid phase transition of atmospheric aerosols. Correlation analysis between SOR and AWC further backs this point up (Fig. R1 in this response, which has been added to the revised SI as Fig. S3, Page 6). In the case of $O_3$ concentration threshold, this is consistent with the consumption of liquid oxidants in multiphase sulfate formation.

2) We agree with the referee that lack of $H_2O_2$ measurement is a weakness in the discussion of possible role of $H_2O_2$ in sulfate formation mechanisms. To add more confidence in such discussion, a proxy measurement of $H_2O_2$ is included in the revised manuscript. Taking the advice of referee #1 (comment NO.4), that $H_2O_2$ was non-linearly correlated with temperature (Fu, 2014). $H_2O_2$ was estimated from temperature, by assuming the same relationship applicable to our measurements in the full year of

2012–2013. As shown in Fig.S2 in this response (added in the revised SI as Fig. S6, Page 9), maximum concentration of $H_2O_2$ in summer is expected and confirmed, which is in line with the fastest sulfate formation in summer all over the year. SOR was further plotted against $H_2O_2$ and positive correlation was found between them (Fig. R3 in this response, which has been added in the revised SI as Fig.S7, Page 9. Please also refer to comment NO.4). In addition, coincident increases in the concentration of $H_2O_2$ and $PM_{2.5}$ in winter of Beijing also lead to an important role of the $H_2O_2$ route in sulfate formation (Ye et al., 2018). These discussions were added up to our previous analysis in the original manuscript, i.e., $O_3$ and $H_2O_2$ are proposed to be the major oxidants in multiphase sulfate formation based on the above threshold analysis. Since $O_3$ was excluded as a major oxidant in multiphase sulfate formation, for that the high aerosol acidity in urban environments limits its reaction rate, $H_2O_2$ remains the only possible liquid phase oxidant (Page 7 lines 14–24 in the revised manuscript). Based on all the above discussions, we carefully proposed in the revised manuscript that $H_2O_2$ might be an important oxidant of sulfate formation.

3) As reminded by referee #1, we double-checked the relationship between SOR and AWC (Fig. R1 in this response, which has been added in the revised SI as Fig. S3, Page 6), and positive correlation between them was found, which further supports that the multiphase reactions, rather than gas phase reactions, are responsible for sulfate formation.

4) The possible role of gas phase reactions was further discussed in the revised manuscript. First, the thresholds of $O_3$ and RH are suggestive of multiphase reactions, as stated above, rather than gas phase reactions, to account for sulfate formation. Second, coincident increases in SOR with aerosol loading (Fig.11 in the revised manuscript, Page 26), with concomitant suppression of photochemistry due to light shielding by aerosols (Wang et al., 2017) and NO-titration of $O_3$ (Page 6 line 19 in the revised manuscript), excludes gas phase reactions as a major route of sulfate formation in Beijing. Last but not the least, gas phase reactions may contribute but are not the major route of sulfate formation, either in Beijing or globally, due to the relatively slow reaction of $SO_2$ with OH. For example, the lifetime of $SO_2$ with respect to OH oxidation

is about 3–4 days, assuming a 24-h average OH concentration of $1 \times 10^6$ molecules cm$^{-3}$ and a pseudo-secondary-order rate constant of $10^{-12}$ cm$^3$ molecules$^{-1}$ s$^{-1}$ (Brothers et al., 2010). However, the overall oxidation lifetime of SO$_2$ is on the order of hours (Berglen et al., 2004; He et al., 2018). Hence, that gas phase reactions contribute but are not the major route of sulfate formation is a well-accepted point in the literature (Finlayson-Pitts and Pitts, 2000; He et al., 2018).

However, we agree with the reviewer that gas phase reactions cannot be neglected and that the gas phase reaction competes with multiphase reactions in sulfate formation. For example, both O$_3$ and RH/water vapor concentration increased in summer with pollution accumulation. As the precursors of OH radicals, the increasing trends of both O$_3$ and water vapor might indicate increasing concentration of OH, and hence reaction rate of SO$_2$ and OH. A discussion of the possible role of gas phase reactions has been added to Page 9 lines 14–20 in the revised manuscript.

[Figure]

**Figure R1.** Plot of the sulfur oxidation ratio (SOR) against aerosol water content (AWC) (note log scale), grouped by O$_3$ concentration. The solid blue circles represent O$_3$ > 35 ppb and the solid black circles represent O$_3$ < 35 ppb. The boxes represent, from top to bottom, the 75$^{th}$, 50$^{th}$, and 25$^{th}$ percentiles in each bin, which were also separated according to the 35 ppb O$_3$ concentration threshold; the bin widths were set such that there were an approximately equal number of data points in each bin. The whiskers, solid squares, and open circles represent 1.5 times the interquartile range (IQR), mean values, and outlier data points, respectively. The lines are best fits to the mean values based on a sigmoid function. Data for days with rain or snow were excluded from this plot.

**Changes in Manuscript:** As for the discussion on $H_2O_2$ oxidation, please refer to the revised manuscript, Page 5 lines 25–27 and Page 7 lines 12–24. For the discussion on gas reaction, please refer to the revised manuscript, Page 9 lines 14–20.

**Comment NO.2:** *The fact of no correlation between SOR and $NO_2$ could make a good argument on the role of $NO_2$ in sulfate formation, I suggest to emphases this point. In addition, comparing SOR, $NO_2$ and $NH_4^+$ (it would be better if $NH_3$ is available), and see if there is any clue on the role of $NH_3$ in aerosol pH and the promoted $NO_2$ oxidation route as proposed by earlier studies.?*

**Response:**

We took the advice and further discussed the possible role of $NO_2+O_2$ route in the revised manuscript based on the following two points. First, no correlation between the SOR and $NO_2$ was found. Secondly, although in our study, $NH_3$ measurements were not available, previous studies have reported a mean aerosol pH value of 4.2 with a low limit of 3.0 in Beijing(Ding et al., 2019; Liu et al., 2017), which suggests that several pH-sensitive routes of sulfate formation, such as $NO_2 + O_2$, TMIs + $O_2$, $O_3$ etc., are highly suppressed. Therefore, we proposed that $NO_2+O_2$ might not be a major mechanism of sulfate formation.

**Changes in Manuscript:** Please refer to the revised manuscript, Page 7 lines 30–32 and Page 8 lines 1–3.

**Comment NO.3:** *It looks the authors dealt with SOR as a sole local phenomenon (local emission and local oxidation), but how about the difference in the regional transport of $SO_2$ and $SO_4^{2-}$? What would this do to SOR?*

**Response:**

Yes, regional transport or intrusion of $SO_2$ and $SO_4^{2-}$ into Beijing has been evidenced in the literature (Lang et al., 2013; Li et al., 2016), and would contributes to SOR. However, our analysis was based on stationary measurements and regional transport could not be considered based on the data we have. Even though, strong relationships between SORs and RH/$O_3$ were still found, revealing the dominant role of Local

multiphase reactions in sulfate formation. Further chemical-transport model study in the future is encouraged to more accurately evaluate the contribution of local chemical formation to sulfate.

**Changes in Manuscript:** Uncertainty analysis introduced from neglecting regional transport has been added to the revised manuscript, Page 2 lines 23–25.

**Comment NO.4:** *There is observational data on the relationship of $H_2O_2$ concentration and temperature in Beijing (Fu, A.: Study on peroxides concentration and its influencing factors in the urban atmosphere, master of engineering, College of Environmental and Resource Sciences, Zhejiang University, Hangzhou, China, 56 pp., 2014 (in Chinese)), the authors can derive the $H_2O_2$ concentration from the temperature data to better constrain the role of $H_2O_2$ by comparisons with $O_3$ and SOR data.*

**Response:** Accepted

According to Fu (2014), $H_2O_2$ was non-linearly correlated with temperature. By assuming the same relationship applicable to our measurements in the full year of 2012–2013, $H_2O_2$ was estimated from temperature and shown in Fig. R2 in this response (added to the revised SI as Fig. S6, Page 9). Maximum concentration of $H_2O_2$ in summer is expected and confirmed, which is in line with the fastest sulfate formation in summer all over the year.

[Figure]

**Figure R2**. Time series of estimated $H_2O_2$ from from March 12012 to February 28 2013 (open black circles). $H_2O_2$ was estimated from temperature (T) based on the fitting function $H_2O_2 = 0.1155e^{0.0846T}$ according to Fu (2014). The boxes represent, from top to bottom, the 75th, 50th, and 25th percentiles for each season. The whiskers, solid red squares, and open red circles represent 1.5 times the interquartile range (IQR), seasonal mean values, and outlier data points, respectively.

SOR was further plotted against $H_2O_2$ and positive correlation was found between them

(Fig. R3 in this response, which has been added to the revised SI as Fig. S7, Page 9), provides more confidence in our discussion of possible role of $H_2O_2$ oxidation in sulfate formation.

[Figure]

**Figure R3**. Plot of the SOR against estimated $H_2O_2$ grouped by RH. The solid blue circles represent RH > 45 % and the solid black circles represent RH < 45 %. The boxes represent, from top to bottom, the 75th, 50th, and 25th percentiles in each bin. The bin widths were set such that there were an approximately equal number of data points in each bin. The whiskers, solid squares, and open circles represent 1.5 times the IQR, mean values, and outlier data points, respectively. The line are best fits to the mean values based on an exponential function. Data for days with rain were excluded from this plot.

**Changes in Manuscript:** The proxy measurement of $H_2O_2$ and further discussion have been added into our revised manuscript, Page 7 lines 14–24.

**Comment NO.5:** *Atmospheric oxidation capacity is a rather vague (or big) definition when related to specific oxidation route of chemicals. Try to avoid*

**Response:** Accepted.

**Changes in Manuscript:** Atmospheric oxidative capacity was replaced by the appropriate oxidants. Please refer to the revised manuscript, Page 6 line 16, Page 8 line 31, Page 9 lines 23–24, and Page 10 line 8.

**Comment NO.6:** *The manuscript need a little bit more tuned, e.g., line 31-32: what is "a given RH threshold"?*

**Response:** Accepted.

A given RH threshold" refers to RH threshold of around 45% observed in our study.

**Changes in Manuscript:** We have rewrite the sentence to "when RH was above a threshold of 45%", please refer to the revised manuscript, Page 5 line 24.

**References**

Berglen, T. F., Berntsen, T. K., Isaksen, I. S. A., and Sundet, J. K.: A global model of the coupled sulfur/oxidant chemistry in the troposphere: The sulfur cycle, J. Geophys. Res. Atmos., 109, https://doi.org/10.1029/2003jd003948, 2004.

Brothers, L. A., Dominguez, G., Abramian, A., Corbin, A., Bluen, B., and Thiemens, M. H.: Optimized low-level liquid scintillation spectroscopy of S-35 for atmospheric and biogeochemical chemistry applications, Proc. Natl. Acad. Sci. U.S.A., 107, 5311-5316, https://doi.org/10.1073/pnas.0901168107, 2010.

Ding, J., Zhao, P., Su, J., Dong, Q., Du, X., and Zhang, Y.: Aerosol pH and its driving factors in Beijing, Atmos. Chem. Phys., 19, 7939-7954, https://doi.org/10.5194/acp-19-7939-2019, 2019.

Finlayson-Pitts, B. J., and Pitts, J. N. Jr.: Chemistry of the upper and lower atmosphere: Theory, experiments, and applications, Academic Press, San Diego, Califoria, 2000.

Fu, A.: Study on peroxide concentration and its influence factors in the urban atmosphere, Master, College of Environmental and Resource Sciences, Zhejiang University, Hangzhou, China, 2014 (in Chinese).

He, P., Alexander, B., Geng, L., Chi, X., Fan, S., Zhan, H., Kang, H., Zheng, G., Cheng, Y., Su, H., Liu, C., and Xie, Z.: Isotopic constraints on heterogeneous sulfate production in Beijing haze, Atmos. Chem. Phys., 18, 5515-5528, https://doi.org/10.5194/acp-18-5515-2018, 2018.

Lang, J. L., Cheng, S. Y., Li, J. B., Chen, D. S., Zhou, Y., Wei, X., Han, L. H., and Wang, H. Y.: A Monitoring and modeling study to investigate regional transport and characteristics of $PM_{2.5}$ pollution, Aerosol Air Qual. Res., 13, 943-956, https://doi.org/10.4209/aaqr.2012.09.0242, 2013.

Li, Y. R., Ye, C. X., Liu, J., Zhu, Y., Wang, J. X., Tan, Z. Q., Lin, W. L., Zeng, L. M., and Zhu, T.: Observation of regional air pollutant transport between the megacity Beijing and the North China Plain, Atmos. Chem. Phys., 16, 14265-14283, https://doi.org/10.5194/acp-16-14265-2016, 2016.

Liu, M., Song, Y., Zhou, T., Xu, Z., Yan, C., Zheng, M., Wu, Z., Hu, M., Wu, Y., and Zhu, T.: Fine particle pH during severe haze episodes in northern China, Geophys. Res. Lett., 44, 5213-5221, https://doi.org/10.1002/2017GL073210, 2017.

Wang, R., Xu, X., Jia, S., Ma, R., Ran, L., Deng, Z., Lin, W., Wang, Y., and Ma, Z.: Lower tropospheric distributions of $O_3$ and aerosol over Raoyang, a rural site in the North China Plain, Atmos. Chem. Phys., 17, 3891-3903, https://doi.org/10.5194/acp-17-3891-2017, 2017.

Ye, C., Liu, P., Ma, Z., Xue, C., Zhang, C., Zhang, Y., Liu, J., Liu, C., Sun, X., and Mu, Y.: High $H_2O_2$ concentrations observed during haze periods in wintertime of Beijing: Importance of $H_2O_2$-oxidation in sulfate formation, Environ. Sci. Technol. Lett., https://doi.org/10.1021/acs.estlett.8b00579, 2018.

---

## Author Response (AR1)

**Response to the Comments of Referees**

**RH and O₃ concentration as two prerequisites for sulfate formation**

Yanhua Fang and Chunxiang Ye, Junxia Wang, Yusheng Wu, Min Hu, Weili Lin, Fanfan Xu, Tong Zhu

We thank the referees for the critical comments, which are very helpful in improving the quality of the manuscript. We have made major revision based on the critical comments and suggestions of the referees. Our point-by-point responses to the comments are listed in the following.

**Anonymous Referee #1**

**Comment NO.1:** *The manuscript by Fang et al. provides a nice year-long dataset of PM$_{2.5}$ along with chemical composition and some important precursors, which would be of interest in improving the understanding of pollution evolution in Beijing. Throughout the manuscript, the authors focused mostly on the observed relationships between SOR and O₃/RH, and made conclusions that O₃ and RH are two "prerequisites" of sulfate formation. These conclusions, however, are predictable. RH and O₃ together provide almost all the necessary conditions for sulfate formation: for gas phase oxidation, they are sources of OH, and for aqueous phase or heterogeneous phase oxidations, water and oxidants (O₃, H₂O₂ (O₃ was a precursor of H₂O₂)).*

*This is saying, that the authors focused on the relationship between SOR and O₃/RH and concluded on multi-phase reaction by H₂O₂ oxidation dominate (or major) sulfate formation is over concluded, or more like a speculation, especially given the absence of H₂O₂ data.*

*In addition, there should be seasonal difference on the formation route, for example, in summer, pollution was the lowest and SOR was the highest, given the data presented, one cannot judge that multi-phase reaction by H₂O₂ oxidation should be responsible for sulfate formation: won't the gas-phase oxidation also enhanced in summer? In fact,*

*for multiphase reactions, AWC might be a better indicator, however, as shown in Figure 7, SOR is not well correlated with AWC but better with RH. This for me is a good if not strong indicator that gas-phase oxidation (promoted by high $O_3$ + RH + insulation) is important for at least summer high SOR.*

**Response:**

We are grateful to the reviewer for the positive and encouraging comments on the dataset and the scientific contribution of our manuscript to understanding sulfate formation.

1) We would like to first summary the main contribution of our manuscript here. Our manuscript is the first to introduce the idea that there are some threshold values (or turning points), above which the SOR increases rapidly, for both RH and $O_3$, based on year-long observations. We presented clear observational evidence for these thresholds, best seen in the plot of SOR versus RH and $O_3$ data (Fig. 5 in the revised manuscript, Page 20). The thresholds at roughly 35 ppb $O_3$ and 45% RH are observed. Although such turning point possible varies in different seasons and locations, such thresholds immediately indicate that both RH and $O_3$ are two "prerequisites" for the multiphase formation of sulfate. In the case of the RH threshold, this is consistent with current understanding in the dependence of the multiphase sulfate formation on aerosol water, since RH threshold relates to the semisolid-to-liquid phase transition of atmospheric aerosols. Correlation analysis between SOR and AWC further backs this point up (Fig. R1 in this response, which has been added to the revised SI as Fig. S3, Page 6). In the case of $O_3$ concentration threshold, this is consistent with the consumption of liquid oxidants in multiphase sulfate formation.

2) We agree with the referee that lack of $H_2O_2$ measurement is a weakness in the discussion of possible role of $H_2O_2$ in sulfate formation mechanisms. To add more confidence in such discussion, a proxy measurement of $H_2O_2$ is included in the revised manuscript. Taking the advice of referee #1 (comment NO.4), that $H_2O_2$ was non-linearly correlated with temperature (Fu, 2014). $H_2O_2$ was estimated from temperature, by assuming the same relationship applicable to our measurements in the full year of

2012–2013. As shown in Fig.S2 in this response (added in the revised SI as Fig. S6, Page 9), maximum concentration of $H_2O_2$ in summer is expected and confirmed, which is in line with the fastest sulfate formation in summer all over the year. SOR was further plotted against $H_2O_2$ and positive correlation was found between them (Fig. R3 in this response, which has been added in the revised SI as Fig.S7, Page 9. Please also refer to comment NO.4). In addition, coincident increases in the concentration of $H_2O_2$ and $PM_{2.5}$ in winter of Beijing also lead to an important role of the $H_2O_2$ route in sulfate formation (Ye et al., 2018). These discussions were added up to our previous analysis in the original manuscript, i.e., $O_3$ and $H_2O_2$ are proposed to be the major oxidants in multiphase sulfate formation based on the above threshold analysis. Since $O_3$ was excluded as a major oxidant in multiphase sulfate formation, for that the high aerosol acidity in urban environments limits its reaction rate, $H_2O_2$ remains the only possible liquid phase oxidant (Page 7 lines 14–24 in the revised manuscript). Based on all the above discussions, we carefully proposed in the revised manuscript that $H_2O_2$ might be an important oxidant of sulfate formation.

3) As reminded by referee #1, we double-checked the relationship between SOR and AWC (Fig. R1 in this response, which has been added in the revised SI as Fig. S3, Page 6), and positive correlation between them was found, which further supports that the multiphase reactions, rather than gas phase reactions, are responsible for sulfate formation.

4) The possible role of gas phase reactions was further discussed in the revised manuscript. First, the thresholds of $O_3$ and RH are suggestive of multiphase reactions, as stated above, rather than gas phase reactions, to account for sulfate formation. Second, coincident increases in SOR with aerosol loading (Fig.11 in the revised manuscript, Page 26), with concomitant suppression of photochemistry due to light shielding by aerosols (Wang et al., 2017) and NO-titration of $O_3$ (Page 6 line 19 in the revised manuscript), excludes gas phase reactions as a major route of sulfate formation in Beijing. Last but not the least, gas phase reactions may contribute but are not the major route of sulfate formation, either in Beijing or globally, due to the relatively slow reaction of $SO_2$ with OH. For example, the lifetime of $SO_2$ with respect to OH oxidation

is about 3–4 days, assuming a 24-h average OH concentration of $1 \times 10^6$ molecules cm$^{-3}$ and a pseudo-secondary-order rate constant of $10^{-12}$ cm$^3$ molecules$^{-1}$ s$^{-1}$ (Brothers et al., 2010). However, the overall oxidation lifetime of SO$_2$ is on the order of hours (Berglen et al., 2004; He et al., 2018). Hence, that gas phase reactions contribute but are not the major route of sulfate formation is a well-accepted point in the literature (Finlayson-Pitts and Pitts, 2000; He et al., 2018).

However, we agree with the reviewer that gas phase reactions cannot be neglected and that the gas phase reaction competes with multiphase reactions in sulfate formation. For example, both O$_3$ and RH/water vapor concentration increased in summer with pollution accumulation. As the precursors of OH radicals, the increasing trends of both O$_3$ and water vapor might indicate increasing concentration of OH, and hence reaction rate of SO$_2$ and OH. A discussion of the possible role of gas phase reactions has been added to Page 9 lines 14–20 in the revised manuscript.

[Figure]

**Figure R1**. Plot of the sulfur oxidation ratio (SOR) against aerosol water content (AWC) (note log scale), grouped by O$_3$ concentration. The solid blue circles represent O$_3$ > 35 ppb and the solid black circles represent O$_3$ < 35 ppb. The boxes represent, from top to bottom, the 75th, 50th, and 25th percentiles in each bin, which were also separated according to the 35 ppb O$_3$ concentration threshold; the bin widths were set such that there were an approximately equal number of data points in each bin. The whiskers, solid squares, and open circles represent 1.5 times the interquartile range (IQR), mean values, and outlier data points, respectively. The lines are best fits to the mean values based on a sigmoid function. Data for days with rain or snow were excluded from this plot.

**Changes in Manuscript:** As for the discussion on $H_2O_2$ oxidation, please refer to the revised manuscript, Page 5 lines 25–27 and Page 7 lines 12–24. For the discussion on gas reaction, please refer to the revised manuscript, Page 9 lines 14–20.

**Comment NO.2:** *The fact of no correlation between SOR and $NO_2$ could make a good argument on the role of $NO_2$ in sulfate formation, I suggest to emphases this point. In addition, comparing SOR, $NO_2$ and $NH_4^+$ (it would be better if $NH_3$ is available), and see if there is any clue on the role of $NH_3$ in aerosol pH and the promoted $NO_2$ oxidation route as proposed by earlier studies.?*

**Response:**

We took the advice and further discussed the possible role of $NO_2+O_2$ route in the revised manuscript based on the following two points. First, no correlation between the SOR and $NO_2$ was found. Secondly, although in our study, $NH_3$ measurements were not available, previous studies have reported a mean aerosol pH value of 4.2 with a low limit of 3.0 in Beijing(Ding et al., 2019; Liu et al., 2017), which suggests that several pH-sensitive routes of sulfate formation, such as $NO_2 + O_2$, TMIs $+ O_2$, $O_3$ etc., are highly suppressed. Therefore, we proposed that $NO_2+O_2$ might not be a major mechanism of sulfate formation.

**Changes in Manuscript:** Please refer to the revised manuscript, Page 7 lines 30–32 and Page 8 lines 1–3.

**Comment NO.3:** *It looks the authors dealt with SOR as a sole local phenomenon (local emission and local oxidation), but how about the difference in the regional transport of $SO_2$ and $SO_4^{2-}$? What would this do to SOR?*

**Response:**

Yes, regional transport or intrusion of $SO_2$ and $SO_4^{2-}$ into Beijing has been evidenced in the literature (Lang et al., 2013; Li et al., 2016), and would contributes to SOR. However, our analysis was based on stationary measurements and regional transport could not be considered based on the data we have. Even though, strong relationships between SORs and RH/$O_3$ were still found, revealing the dominant role of Local

multiphase reactions in sulfate formation. Further chemical-transport model study in the future is encouraged to more accurately evaluate the contribution of local chemical formation to sulfate.

**Changes in Manuscript:** Uncertainty analysis introduced from neglecting regional transport has been added to the revised manuscript, Page 2 lines 23–25.

**Comment NO.4:** *There is observational data on the relationship of $H_2O_2$ concentration and temperature in Beijing (Fu, A.: Study on peroxides concentration and its influencing factors in the urban atmosphere, master of engineering, College of Environmental and Resource Sciences, Zhejiang University, Hangzhou, China, 56 pp., 2014 (in Chinese)), the authors can derive the $H_2O_2$ concentration from the temperature data to better constrain the role of $H_2O_2$ by comparisons with $O_3$ and SOR data.*

**Response:** Accepted

According to Fu (2014), $H_2O_2$ was non-linearly correlated with temperature. By assuming the same relationship applicable to our measurements in the full year of 2012–2013, $H_2O_2$ was estimated from temperature and shown in Fig. R2 in this response (added to the revised SI as Fig. S6, Page 9). Maximum concentration of $H_2O_2$ in summer is expected and confirmed, which is in line with the fastest sulfate formation in summer all over the year.

[Figure]

**Figure R2**. Time series of estimated $H_2O_2$ from from March 12012 to February 28 2013 (open black circles). $H_2O_2$ was estimated from temperature (T) based on the fitting function $H_2O_2 = 0.1155e^{0.0846T}$ according to Fu (2014). The boxes represent, from top to bottom, the 75[th], 50[th], and 25[th] percentiles for each season. The whiskers, solid red squares, and open red circles represent 1.5 times the interquartile range (IQR), seasonal mean values, and outlier data points, respectively.

SOR was further plotted against $H_2O_2$ and positive correlation was found between them

(Fig. R3 in this response, which has been added to the revised SI as Fig. S7, Page 9), provides more confidence in our discussion of possible role of $H_2O_2$ oxidation in sulfate formation.

[Figure]

**Figure R3**. Plot of the SOR against estimated $H_2O_2$ grouped by RH. The solid blue circles represent RH > 45 % and the solid black circles represent RH < 45 %. The boxes represent, from top to bottom, the 75[th], 50[th], and 25[th] percentiles in each bin. The bin widths were set such that there were an approximately equal number of data points in each bin. The whiskers, solid squares, and open circles represent 1.5 times the IQR, mean values, and outlier data points, respectively. The line are best fits to the mean values based on an exponential function. Data for days with rain were excluded from this plot.

**Changes in Manuscript:** The proxy measurement of $H_2O_2$ and further discussion have been added into our revised manuscript, Page 7 lines 14–24.

**Comment NO.5:** *Atmospheric oxidation capacity is a rather vague (or big) definition when related to specific oxidation route of chemicals. Try to avoid*

**Response:** Accepted.

**Changes in Manuscript:** Atmospheric oxidative capacity was replaced by the appropriate oxidants. Please refer to the revised manuscript, Page 6 line 16, Page 8 line 31, Page 9 lines 23–24, and Page 10 line 8.

**Comment NO.6:** *The manuscript need a little bit more tuned, e.g., line 31-32: what is "a given RH threshold"?*

**Response:** Accepted.

A given RH threshold" refers to RH threshold of around 45% observed in our study.

**Changes in Manuscript:** We have rewrite the sentence to "when RH was above a threshold of 45%", please refer to the revised manuscript, Page 5 line 24.

(or turning points), above which the SOR increases rapidly, for both RH and $O_3$, based on year-long observations. We presented clear observational evidence for these thresholds, best seen in the plot of SOR versus RH and $O_3$ data (Fig. 5 in the revised manuscript, Page 20). The thresholds at roughly 35 ppb $O_3$ and 45% RH are observed. Although such turning point possible varies in different seasons and locations, such thresholds immediately indicate that both RH and $O_3$ are two "prerequisites" for the multiphase formation of sulfate. In the case of the RH threshold, this is consistent with current understanding in the dependence of the multiphase sulfate formation on aerosol water, since RH threshold relates to the semisolid-to-liquid phase transition of atmospheric aerosols. Correlation analysis between SOR and AWC further backs this point up (Fig. R1 in this response, which has been added to the revised SI as Fig. S3, Page 6). In the case of $O_3$ concentration threshold, this is consistent with the consumption of liquid oxidants in multiphase sulfate formation.

[Figure]

**Figure R1**. Plot of the sulfur oxidation ratio (SOR) against aerosol water content (AWC) (note log scale), grouped by $O_3$ concentration. The solid blue circles represent $O_3 > 35$ ppb and the solid black circles represent $O_3 < 35$ ppb. The boxes represent, from top to bottom, the 75th, 50th, and 25th percentiles in each bin, which were also separated according to the 35 ppb $O_3$ concentration threshold; the bin widths were set such that there were an approximately equal number of data points in each bin. The whiskers, solid squares, and open circles represent 1.5 times the interquartile range (IQR), mean values, and outlier

data points, respectively. The lines are best fits to the mean values based on a sigmoid function. Data for days with rain or snow were excluded from this plot.

2) We agree with the referee that lack of $H_2O_2$ measurement is a weakness in the discussion of possible role of $H_2O_2$ in sulfate formation mechanisms. To add more confidence in such discussion, a proxy measurement of $H_2O_2$ is included in the revised manuscript. Taking the advice of referee #1, that $H_2O_2$ was non-linearly correlated with temperature (Fu, 2014). $H_2O_2$ was estimated from temperature, by assuming the same relationship applicable to our measurements in the full year of 2012–2013. As shown in Fig.S2 in this response (added in the revised SI as Fig. S6, Page 9), maximum concentration of $H_2O_2$ in summer is expected and confirmed, which is in line with the fastest sulfate formation in summer all over the year. SOR was further plotted against $H_2O_2$ and positive correlation was found between them (Fig. R3 in this response, which has been added in the revised SI as Fig.S7, Page 9.). In addition, coincident increases in the concentration of $H_2O_2$ and $PM_{2.5}$ in winter of Beijing also lead to an important role of the $H_2O_2$ route in sulfate formation (Ye et al., 2018). These discussions were added up to our previous analysis in the original manuscript, i.e., $O_3$ and $H_2O_2$ are proposed to be the major oxidants in multiphase sulfate formation based on the above threshold analysis. Since $O_3$ was excluded as a major oxidant in multiphase sulfate formation, for that the high aerosol acidity in urban environments limits its reaction rate, $H_2O_2$ remains the only possible liquid phase oxidant (Page 7 lines 14–24 in the revised manuscript). Based on all the above discussions, we carefully proposed in the revised manuscript that $H_2O_2$ might be an important oxidant of sulfate formation.

[Figure]

**Figure R2**. Time series of estimated $H_2O_2$ from from March 12012 to February 28 2013 (open black circles). $H_2O_2$ was estimated from temperature (T) based on the fitting function $H_2O_2 = 0.1155e^{0.0846T}$ according to Fu (2014). The boxes represent, from top to bottom, the 75th, 50th, and 25th percentiles for each season. The whiskers, solid red squares, and open red circles represent 1.5 times the interquartile range (IQR), seasonal mean values, and outlier data points, respectively.

[Figure]

**Figure R3**. Plot of the SOR against estimated $H_2O_2$ grouped by RH. The solid blue circles represent RH > 45 % and the solid black circles represent RH < 45 %. The boxes represent, from top to bottom, the 75th, 50th, and 25th percentiles in each bin. The bin widths were set such that there were an approximately equal number of data points in each bin. The whiskers, solid squares, and open circles represent 1.5 times the IQR, mean values, and outlier data points, respectively. The line are best fits to the mean values based on an exponential function. Data for days with rain were excluded from this plot.

**Changes in Manuscript:** A summary of our scientific contribution has been revised in the abstract and in the text, please refer to the revised manuscript, Page 1 lines 13–19 and Page 5 lines 25–26. Further discussions on the role of $H_2O_2$ has also been added to the revised manuscript, Page 7 lines 14–24.

**Comment NO.3:** *The author name should be Weili Lin.*

**Response:** Accepted.

**Changes in Manuscript:** We have made a correction, please refer to the revised manuscript, Page 1 line 2.

**Comment NO.4:** *"threshold of RH and ozone" Where is this statement coming from? Is it a definition/estimate of the authors? If the threshold changed with different locations and seasons? What is the effect of these thresholds?*

**Response:**

1) "Thresholds of RH and ozone" are obtained based our measurement in the full year of 2012-2013 that above some turning points of RH and $O_3$ concentration, SORs increase rapidly. This is best seen in the plot of SOR versus RH and $O_3$ data (Fig. 5 in the original manuscript, Page 20). Our interpretation of this is that there are thresholds or turning points in RH and $O_3$ concentration that must be exceeded to allow for the fast formation of sulfate. Although such turning point possible varies in different seasons and locations, such thresholds immediately indicate that both RH and $O_3$ are two "prerequisites" for the multiphase formation of sulfate.

2) It is also the authors' interpretation that the threshold of RH is around 45 % and the threshold of $O_3$ is around 35 ppb. There could be some uncertainty attached with such inferred values. For example, one could argue that the threshold of $O_3$ concentration is any value between 30–40 ppb. Also, the daily average RH and $O_3$ data used in our analyses are not the best to evaluate the thresholds. For example, the observed RH threshold is proposed to be determined by the phase transition RH. However, the timescale of the phase transition in ambient air is on the order of seconds (Liu et al., 2008), in comparison to RH changes on timescales of hours to days, and thus the daily average RH is not an accurate estimate of the phase transition RH. This explains why the apparent RH threshold of 45 % observed in Fig. 5 is somewhat below the *in situ* phase transition RH of 50–60 % (Liu et al., 2017b).

3) The thresholds might change with locations and seasons. For instance, Fig. R4 in this response (added to the revised manuscript as Fig. 6, Page 21) suggests that the RH threshold is roughly around 45 % during all four seasons in Beijing. The turning point varied within 40%- 50% in different sampling location of Beijing (Liu et al., 2015; Xu et al., 2017; Yang et al., 2015; Zheng et al., 2015). However, similar analyses must be performed using high time resolution data to confirm the trends observed based on our daily average data.

[Figure]

**Figure R4**. Plots of SORs against RH, grouped by $O_3$ concentration in four seasons. The solid blue circles represent $O_3 > 35$ ppb and the solid black circles represent $O_3 < 35$ ppb. The boxes represent, from top to bottom, the $75^{th}$, $50^{th}$, and $25^{th}$ percentiles in each bin ($\Delta RH = 5$ %). The whiskers, solid red squares, and open red circles represent 1.5 times the IQR, mean values, and outlier data points, respectively. The red lines are best fits to mean values based on sigmoid function. Data for days with rain or snow were excluded from these plots.

4)  As stated above, above the thresholds of RH and $O_3$ concentration, sulfate formation could be enhanced (Please also refer to the response of comment NO.2).

**Changes in the Manuscript:** A discussion on the possible seasonal variations in the thresholds were added in our revised manuscript, please refer to the revised manuscript, Page 6 lines 32–34 and Page 7 lines 1–7.

**Comment NO.5:** *Redundancy: Page 1 line 15-16 and line 24-25. Line 13-14 and Line 17-18.*

**Response:** Accepted.

**Changes in the Manuscript:** We have rewritten the abstract and deleted the redundant sentences in the revised manuscript. Please refer to the revised manuscript, Page 1 lines 14–28.

**Comment NO.6:** *Section 2.1.2. Please add the steps of weighing after sampling.*

**Response:**

The steps of weighting after sampling have been provided in the original manuscript. Please refer to the revised manuscript, Page 4 lines 3–5 (highlighted).

**Comment NO.7:** *Page 4, line 27. Should be annual standard*

**Response:** Accepted.

**Changes in Manuscript:** We have changed the phrase to "Chinese National Ambient Air Standard annual mean concentration of ", please refer to the revised manuscript, Page 5 line 5.

**Comment NO.8:** *Page 5, line 2. The method to calculate POM should be introduced in previous section.*

**Response:**

The method to calculate POM was provided in the original SI. The discussion on source appointment, including POM, has been deleted in the revised manuscript and SI.

**Comment NO.9:** *Overall, section 3.1 is not necessary, because it has nothing to do with the main idea. If this section is deleted in the main article, it will not affect the presentation of the article. For example, the authors described the measurements of ions, organics and metal. However, ions except SNA, organics and metals except Fe didn't help the discussion of your topic. Therefore, the method and results section should to be streamlined.*

**Response:** Accepted

**Changes in Manuscript:** Sect 3.1 has been reduced so that a general description of data is presented, and that variations in $PM_{2.5}$ and its main components are introduced. Please refer to the revised manuscript, Page 5 lines 3–18.

**Comment NO.10:** *Section 3.2. I strongly recommend the authors discussing the relationship between sulfate and RH/ozone in different seasons. The threshold should*

*be changed with seasons.*

**Response:** Accepted

**Changes in Manuscript:** The seasonal variations are discussed now in the revised manuscript (also refer to response to comment NO.4). Please refer to the revised manuscript, Page 6 lines 32–34 and Page 7 lines 1–7.

**Comment NO.11:** *Page 7, line 12-16 repeats the previous statement.*

**Response:**

We intended to summarise our major findings and discuss their implications in this section.

**Changes in Manuscript:** We have rewritten the sentences, please refer to the revised manuscript, Page 7 lines 12–24.

**Comment NO.12:** *Page 7, line 14. What is the atmospheric oxidative capacity? From your statement, does ozone concentration correspond to this? Is it correct? Do you have some references to support your opinion? The authors should clarify this question because the same definition is also used in Page 9, line 20.*

**Response:** Accepted.

Atmospheric oxidative capacity relates to the concentrations of major oxidants such as OH radicals, $O_3$, etc. (Murray et al., 2009). Since $O_3$ is a major oxidant and a precursor to other major oxidants, including OH radicals, to a certain degree, $O_3$ can be used as a proxy for atmospheric oxidative capacity. To improve clarity, atmospheric oxidative capacity was replaced by the appropriate oxidant in each context in the revised manuscript.

**Changes in Manuscript:** Atmospheric oxidative capacity was replaced by the appropriate oxidant. Please refer to the revised manuscript, Page 6 line 16, Page 8 line 31, Page 9 lines 23–24, and Page 10 line 8.

**Comment NO.13:** *Page 7, Line 23-24. Since you couldn't exclude NO₂-based reactions*

*as major route of sulfate formation, the analysis of the relationship between SOR and NO₂ is not necessary.*

**Response:**

We took the advice of referee #1 and further discussed the possible role of $NO_2+O_2$ route in the revised manuscript based on two points. First, no correlation between the SOR and $NO_2$ was found. Secondly, although in our study, $NH_3$ measurements were not available, previous studies has reported a mean aerosol pH value of ~4.2 with a low limit of ~3.0 in Beijing (Ding et al., 2019; Liu et al., 2017a), which suggests that several routes of sulfate formation, such as $NO_2 + O_2$, TMIs $+ O_2$, $O_3$ etc., are suppressed. Therefore, we proposed that $NO_2+O_2$ might not be a major mechanism of sulfate formation.

**Changes in Manuscript:** Please refer to the revised manuscript, Page 7 lines 30–32 and Page 8 lines 1–3.

**Comment NO.14:** *Page 9, line 2-3. The authors described on page 7, line 7-10 that the self-catalytic nature is beyond the scope of your study. However, you illustrate the importance of the self–catalytic in this paragraph. I think it's self-contradictory.*

**Response:**

To clarify, our manuscript states that the self-constrained nature, i.e., sulfate formation increases the acidity of aerosols, which suppresses sulfate formation via several routes, such the $O_3$ oxidation and TMIs $+ O_2$ routes. The self-catalytic nature of sulfate formation is best seen from the perspective that sulfate formation adds up the aerosol volume/surface density which helps with further sulfate formation. Those two mechanisms compete in determining the sulfate formation as pollution accumulation. In our manuscript, the self-constrained nature of sulfate formation is not discussed in detail due to the lack of direct or proxy measurements of aerosol acidity in our measurements.

**Comment NO.15:** *Page 10, line 21. Should be Zhejiang University.*

**Response:** Accepted.

**Changes in Manuscript:** We have made the correction. Please refer to the revised manuscript, Page 11 lines 18–19.

[Figure]

**Figure R1**. Time series of estimated $H_2O_2$ from from March 12012 to February 28 2013 (open black circles). $H_2O_2$ was estimated from temperature (T) based on the fitting function $H_2O_2 = 0.1155e^{0.0846T}$ according to Fu (2014). The boxes represent, from top to bottom, the 75th, 50th, and 25th percentiles for each season. The whiskers, solid red squares, and open red circles represent 1.5 times the interquartile range (IQR), seasonal mean values, and outlier data points, respectively.

[Figure]

**Figure R2**. Plot of the SOR against estimated $H_2O_2$ grouped by RH. The solid blue circles represent RH > 45 % and the solid black circles represent RH < 45 %. The boxes represent, from top to bottom, the 75th, 50th, and 25th percentiles in each bin. The bin widths were set such that there were an approximately equal number of data points in each bin. The whiskers, solid squares, and open circles represent 1.5 times the IQR, mean values, and outlier data points, respectively. The line are best fits to the mean values based on an exponential function. Data for days with rain were excluded from this plot.

**Changes in Manuscript:** Discussions on the role of $H_2O_2$ has also been added to the revised manuscript, Page 7 lines 14–24.

**Comment NO.2:** *The authors should adjust the structures of the paper to make more clear and concise statement. Although the overview of the data is needed for the readers, the discussion in Sect3.1 is concentrated on the source appointment of $PM_{2.5}$, which is abundant and deviate away from the theme. I suggest this Sect. discuss the variations of the components concentrations and contribution ratios using the classification method based on season or pollution levels. Sulfate can be focused on.*

**Response:** Accepted

**Changes in Manuscript:** Sect 3.1 has been reduced so that a general description of data is presented, and that variations in $PM_{2.5}$ and its main components are introduced. Please refer to the revised manuscript, Page 5 lines 3–18.

**Comment NO.3:** *The order of the figures and tables in the main text and SI is confusing, the authors should rearrange the figures and tables according to the main text.*

**Response:** Accepted

**Changes in Manuscript:** We have rearranged the figures. Please refer to the revised manuscript, Pages 18–19 Figs 3–4.

**Comment NO.4:** *The authors should carefully go through the whole manuscript to avoid mistakes. Specific points: 1. Avoid duplicated sentences and definitions. E.g. Page1, line18- 20 vs Page 2, line 1-2; Page 1, line 25-26 vs Page2, line 23-26, and the definition of "self-catalytic" is vague.*

**Response:** Accepted

1) Duplicated sentences deleted in the revised manuscript.

2) We need to better define the term "self-catalytic" as referee #2 has also suggested. We have therefore defined it consistently in both the abstract and introduction. The definition has changed to: "the formation of hydrophilic sulfate aerosols under high RH conditions results in an increase in aerosol water content, which results in greater particle volume for further multiphase sulfate formation".

**Changes in Manuscript:** The definition has been clarified, please refer to the revised manuscript, Page 1 lines 25–27 and Page 2 lines 16–18.

**Comment NO.5:** *Page 2, line 14, what is "various parameters" refer to*

**Response:** oxidants, catalysts, meteorological conditions, etc.

**Changes in Manuscript:** We have clarified the parameters as "exactly how do various parameters (oxidants, catalysts, meteorological conditions, etc.) influence sulfate formation" in the revised manuscript, Page 2 line 10.

**Comment NO.6:** *Page 4, line 6, Figure 1 should be "Fig. 1"; Page 4, line 15, give the location information (lat, long) of the site; Page 5, line 4-10, rewrite the first sentence "The chemical. . … . (TEOs)." There actually 8 categories including "others" and the category is not according to the source type. Why you start with Fig. S2 not S1? Page 6 why you put Fig. 4 before Fig.3 in your text. Check the orders as mentioned in general points 3.*

**Response:** Accepted.

**Changes in Manuscript:**

1) Figure 1 has been changed to Fig. 1. Please refer to the revised manuscript, Page 3 line 22.

2) The lat/long of the Beijing Meteorological Observatory Station (116.47° E, 39.81° N) has been added. Please refer to the revised manuscript, Page 3 line 29.

3) The sentence the reviewer mentions has been rewritten to: "The chemical components of $PM_{2.5}$ were divided into eight categories: sulfate, nitrate, ammonium, organic matter (OM), EC, minerals, trace element oxides (TEOs), and others." Please refer to the revised manuscript, Page 4 lines 14–15.

4) We have rearranged the order of Figs. Please refer to the revised manuscript, Page 4 lines 17–18 and Pages 18–19 Figs. 3–4.

**Comment NO.7:** *Sect. 3.2 How do you give the definition of threshold? The SOR or ΔSOR exceed certain value? The authors also compared the results with previous studies in this Sect., what is the reason for the difference in these studies?*

**Response:**

5) "Thresholds of RH and ozone" are obtained based our measurement in the full year of 2012-2013 that above some turning points of RH and $O_3$ concentration, SORs increase rapidly. This is best seen in the plot of SOR versus RH and $O_3$ data (Fig. 5 in the original manuscript, Page 20). Our interpretation of this is that there are thresholds or turning points in RH and $O_3$ concentration that must be exceeded to allow for the fast formation of sulfate. Although such turning point possible varies in different seasons and locations, such thresholds immediately indicate that both RH and $O_3$ are two "prerequisites" for the multiphase formation of sulfate.

6) It is also the authors' interpretation that the threshold of RH is around 45 % and the threshold of $O_3$ is around 35 ppb. There could be some uncertainty attached with such inferred values. For example, the thresholds might change with locations and seasons. Also, the daily average RH and $O_3$ data used in our analyses are not the best to evaluate the thresholds. For example, the observed RH threshold is proposed to be determined by the phase transition RH. However, the timescale of the phase

transition in ambient air is on the order of seconds (Liu et al., 2008), in comparison to RH changes on timescales of hours to days, and thus the daily average RH is not an accurate estimate of the phase transition RH. This explains why the apparent RH threshold of 45 % observed in Fig. 5 is somewhat below the *in situ* phase transition RH of 50–60 % (Liu et al., 2017).

**Comment NO.8:** *Page 9, line 5-8 and Page 9, line 12-14 the sentences are contradictory*

**Response:**

The sentences on Page 9, lines 5–8 explain that the self-catalytic nature of sulfate formation accounts for the increased SOR as pollution accumulates. The sentences on page 9, lines 12–14 summarise our conclusion about the thresholds of $O_3$ and RH.

**Comment NO.9:** *Use "clear", "formation", "evolution" etc. to represent different pollution level is improper, because you do not conduct case or course study in the paper.*

**Response:** Accepted.

**Changes in Manuscript:** The definitions have been changed to: clean, moderate pollution, heavy pollution, and severe pollution in the revised manuscript. Please refer to the revised manuscript, Page 9 line4, Page 25 Fig. 10, and Page 26 Fig.11. These still represent each quartile of $PM_{2.5}$ levels.

**Comment NO.10:** *How about other factors such as wind speed and wind direction impact on SOR except RH and $O_3$?*

**Response:**

Wind speed and wind direction are not assumed to be influencing parameters of sulfate formation according to the mechanism summarised in the introduction section and hence were not discussed in our manuscript. However, it is clear that high SORs and high $PM_{2.5}$ were commonly found at low to medium wind speeds (Fig. R3 in this response), which might be related to the increasing SORs as aerosol pollution

accumulated. Hotspots of SOR at high wind speed with northwest sector and south sector are also found, which might be related to regional transport of sulfate. The uncertainty concerning regional transport has been discussed in the response to referee #1 comment NO.3.

[Figure]

**Figure R3**. Bivariate polar plots for (a) SOR and (b) PM$_{2.5}$. The grey shading indicates lack of data. Wind speed and wind direction were download from the National Climate Data Center (www.ncdc.noaa.gov), which were measured at a station located in the Beijing Capital International Airport.

**Comment NO.11:** *Is all the data in this paper daily data? Please give make it clear in the paper.*

**Response:**

Yes, all the data used in this manuscript are daily averages and this has been clarified in the method section of the revised manuscript (Page 3, lines 29–30 in the revised manuscript).

To be more specific, daily PM$_{2.5}$ filter samples were collected for 23.5 h, from 9:30 am to 9:00 am the next day; thus, PM$_{2.5}$ and its components were daily averaged data. Gaseous pollutants (SO$_2$, O$_3$, NO$_x$, etc.) and RH with a time resolution of mins were averaged according to the filter sampling time period. Daily solar radiation data was used as it is.

**Comment NO.12:** *SOR is the conversion ratio of SO$_2$, I doubt whether it can indicate the conversion rate (or speed) as you mentioned in your paper (e.g.   Page 1, line 21, Page 10, line 14 etc.)   What is the relationship of O$_3$ and atmospheric oxidative*

*capacity? AWC and RH? Please reconsider in your statement and discussions? (Page 8, line 10, Page 9, line 10-11 etc.).*

**Response:**

1) We agree with the referee that SOR is defined as the ratio of sulfate to total sulfur and it is not the $SO_2$-sulfate conversion rate. However, due to the long chemical lifetime of sulfate, sulfate is tend to accumulate with chemical production within at least 24 hrs, which could be best reflected in SOR, the ratio of sulfate to total sulfur. SOR has been widely used as an indicator of $SO_2$-to-sulfate conversion in numbers of references (Sun et al., 2014; Zheng et al., 2015), where a high SOR reflects a high $SO_2$-to-sulfate conversion rate on average during the measurement period.

2) Atmospheric oxidative capacity relates to the concentrations of major oxidants such as OH radicals, $O_3$, etc. (Murray et al., 2009). Since $O_3$ is a major oxidant and a precursor to other major oxidants, including OH radicals, to a certain degree, $O_3$ can be used as a proxy for atmospheric oxidative capacity. To improve clarity, atmospheric oxidative capacity was replaced by the appropriate oxidant in each context in the revised manuscript.

3) The AWC calculated using the ISORROPIA-II thermodynamic model (http://isorropia.eas.gatech.edu). Please also refer to the revised SI (Page 3 lines 14–16). In brief, AWC is a function of aerosol mass concentration, aerosol chemical composition, RH, etc.

**Changes in Manuscript:** Atmospheric oxidative capacity was replaced by the appropriate oxidants. Please refer to the revised manuscript, Page 6 line 16, Page 8 line 31, Page 9 lines 23–24, and Page 10 line 8.

**Comment NO.13:** *The fitting methods were used in this paper (Fig. 5 and Fig. S5), please give the evaluation parameters (such as p-value and R) of the fitting method to prove the validity and accuracy of the fitting. Also in Fig 5b, the last 2 box bins only have 1-2 points, does the results make sense?*

**Response:** Accepted

1) $R^2$ has been added to Fig. 5 in the revised manuscript (Page 20).

2) In Fig. 5 (Page 20 in the revised manuscript), $O_3$ concentrations were grouped by 5 ppb intervals and RH by 5 % intervals. There were only a few data points on the right-hand sides of these figures because there were only a few days with daily average $O_3$ (RH) above 70 ppb (70 %). However, the shapes of the fits are not much different when we group them by the number of data points in each bin, as show in Fig. R4 in this response. $O_3$ in Fig. R4a was the original method that grouped by 5 ppb intervals, while $O_3$ in Fig. R4b were grouped with an approximately equal number of data points (15-16) in each bin, which shows the robustness of our fitting.

[Figure]

**Figure R4**. Plots of the SOR against $O_3$, grouped by RH. The solid blue circles represent RH > 45 % and the solid black circles represent RH < 45 %. The boxes represent, from top to bottom, the 75[th], 50[th], and 25[th] percentiles in each bin ((a) $\Delta O_3$ = 5 ppb, (b) variable $\Delta O_3$, 15–16 data points in each bin). The whiskers, solid red squares, and open red circles represent 1.5 times IQR, mean values, and outlier data points, respectively. The red lines are best fits to the mean values based on a sigmoid function. Data for days with rain or snow were excluded from these plots.

**Changes in Manuscript:** $R_2$ has been added to the plots that containing fitting lines. Please refer to the revised manuscript, Page 20 Fig. 5, Page 21 Fig. 6, Page 22 Fig. 7. Please also refer to the revised SI, Page 6 Fig. S3, and Page 9 Fig. S7.

**Comment NO.14:** *Give the right form of the author's name in Page 1 and Page 12. There should be a space between units and the quantity.*

**Response:** Accepted.

**Changes in Manuscript:**

1) The right form of the author's name has been given. Please refer to the revised manuscript, Page 1 line 2.

2) Space has been added between number and % or number between °C, Please refer to the revised manuscript, Page 1 line13, Page 4 lines 6 and 29, Page 5 lines 5, 14, 16, 23, 24, 26, and 29, Page 6 lines 28–29, Page 8 lines 21–22 and 29, Page 9 lines 3–4 and 32, Page 20 lines 4–5 and legend of Fig. 5b, Page 24 line 5, and Page 26 line 6.

[revised manuscript text omitted]

$$EF_i = \frac{[X_i/X_{ref}]_{sample}}{[X_i/X_{ref}]_{crust}}$$ ,

(Eq. 3)

where $[X_i/X_{ref}]_{sample}$ is the mass concentration ratio of element $i$ to the reference element in our samples and $[X_i/X_{ref}]_{crust}$ is the mass concentration ratio of element $i$ to the reference element in average crust (Hans Wedepohl, 1995). Al was used as the reference element in this study. The EFs of each element are depicted in Fig. S2.

[Figure]

**Figure S2.** Elemental enrichment factors (EFs) of our samples. The boxes represent, from top to bottom, the 75[th], 50[th], and 25[th] percentiles for each element. The whiskers, solid red squares, and open red circles represent 1.5 times the interquartile range (IQR), mean values, and outlier data points, respectively.

If the EF was < 5, the element was considered to originate mainly from natural sources; if 5 < EF< 20, the element originated from both natural and anthropogenic sources; if EF > 20, the element originated mainly from anthropogenic sources. According to Zhang et al. (2013), the mass concentrations of TEOs can be estimated by multiplied a correction factor to represent the contribution of oxygen. For elements originating from anthropogenic sources only, a factor of 1 was applied, whereas for elements of both natural and anthropogenic origin, a factor of 0.5 was applied to represent the anthropogenic part. As multiple forms of metal oxides were identified, which were hard to quantify, a multiplicative factor of 1.3 was used when considering the metal abundance. The mass concentration of TEOs was calculated as described in Zhang et al. (2013):

$$[TEOs] = 1.3 \times [0.5 \times (Ba + Mn + U) + (Ni + Co + Cr + Mo + Tl + Cu + Zn + Pb + Cd + Se)] \quad,$$

(Eq. 4)

**S1.4 Aerosol water content**

Aerosol water content (AWC) was calculated using the ISORROPIA-II thermodynamic model (http://isorropia.eas.gatech.edu). The $Na^+$–$K^+$–$Ca^{2+}$–$Mg^{2+}$–$NH_4^+$–$SO_4^{2-}$–$NO_3^-$–$Cl^-$–$H_2O$ aerosol system was applied in reverse mode (Fountoukis and Nenes, 2007; Nenes et al., 1998).

**S2 Results and discussion**

**S2.1 Sulfate formation mechanism**

Sulfate can be formed through the oxidation of $SO_2$ by OH radicals in the gas phase (Stockwell and Calvert, 1983), through the oxidation of dissolved $SO_2$ by various oxidants (e.g., $O_3$, $H_2O_2$, $NO_2$, and $O_2$) in the aqueous phase (Seinfeld and Pandis, 2006), which may be transition metal ions (TMIs)-catalysed, or through heterogeneous reaction on the surface of sea-salt or dust aerosols (Gurciullo et al., 1999; Usher, 2002).

The rate of the $SO_2 + OH$ reaction can be expressed as:

$$R_{SO_2+OH} = k_0[SO_2(g)][OH(g)]$$ ,

(Eq. 5)

where $k_0$ is the rate constant and $[x]$ represents the concentration of species $x$. The production rate of sulfate through OH radical oxidation can be expressed as:

$$P_{OH} = \frac{3600 \times 96 \times p \times R_{SO_2+OH}}{RT}$$ ,

(Eq. 6)

where 3600 is a time conversion factor ($s\ h^{-1}$), 96 is the molar mass of $SO_4^{2-}$ ($g\ mol^{-1}$), $p$ is atmospheric pressure (kPa), $R$ is the gas constant ($8.31\ Pa\ m^3\ mol^{-1}\ K^{-1}$), and $T$ is the temperature (K).

$SO_2$ reacts with $H_2O_2$, $O_3$, $NO_2$, and $O_2$ (TMIs-catalysed) in the aqueous phase. The rates of the four main aqueous reactions are expressed as (He et al., 2018; Seinfeld and Pandis, 2006):

$$R_{SO_2+O_3} = (k_1[SO_2 \cdot H_2O] + k_2[HSO_3^-] + k_3[SO_3^{2-}])[O_3(aq)]$$ ,

(Eq. 7)

$$R_{SO_2+H_2O_2} = \frac{k_4[H^+][HSO_3^-][H_2O_2(aq)]}{1 + K[H^+]}$$ ,

(Eq. 8)

$$R_{SO_2+NO_2} = k_5[S(IV)][NO_2(aq)]$$ ,

(Eq. 9)

$$R_{SO_2+O_2} = k_6[H^+]^{-0.74}[S(IV)][Mn(II)][Fe(III)] \qquad (pH < 4.2)$$ ,

(Eq. 10)

$$R_{SO_2+O_2} = k_7[H^+]^{0.67}[S(IV)][Mn(II)][Fe(III)] \qquad (pH > 4.2)$$ ,

The production rate of sulfate through aqueous oxidation routes can be expressed as:

$$P_{\text{aqu(ox}_i)} = 3600 \times 96 \times R_{\text{SO}_2+\text{ox}_i} \times \frac{\text{LWC}}{\rho_{\text{H}_2\text{O}}}$$

,

(Eq. 12)

where $k_n$ ($n$ = 1–7) is the rate constant of each oxidation route, $K$ = 13 M$^{-1}$ at 298 K, LWC is the liquid water content (mg m$^{-3}$), $\rho_{\text{H2O}}$ is the density of water (1 kg L$^{-1}$), and ox$_i$ ($i$ = O$_3$, H$_2$O$_2$, NO$_2$, and O$_2$) represents different oxidants.

The heterogeneous reaction rate $R_{\text{het(ox}_i)}$ can be expressed as (Jacob, 2000; Wang et al., 2012; Zheng et al., 2015):

$$R_{\text{het(ox}_i)} = k_{\text{ox}_i}[\text{SO}_2(\text{g})]$$ ,

(Eq. 13)

where

$$k_{\text{ox}_i} = \left(\frac{d_\text{p}}{2D_\text{i}} + \frac{4}{v_i\,\gamma_i}\right)^{-1} S_\text{p}$$

,

(Eq. 14)

$d_\text{p}$ is the effective diameter of the particles (m), $D_i$ is the gas phase molecular diffusion coefficient (m$^2$ s$^{-1}$), $v_i$ is the mean molecular speed in the gas phase (m s$^{-1}$), and $S_\text{p}$ is the aerosol surface area (m$^2$ m$^{-3}$).

The uptake coefficient $\gamma_i$ depends on RH:

$$\gamma_i = \begin{cases} \gamma_{\text{low}} & 0 < \text{RH} \leqslant 50\ \% \\ \gamma_{\text{low}} + \dfrac{(\gamma_{\text{high}} - \gamma_{\text{low}})(\text{RH} - 0.5)}{\text{RH}_{\text{max}} - 0.5} & 50\ \% < \text{RH} \leqslant \text{RH}_{\text{max}} \\ \gamma_{\text{high}} & \text{RH}_{\text{max}} < \text{RH} \leqslant 100\ \% \end{cases}$$

(Eq. 15)

where $\gamma_{\text{low}}$ and $\gamma_{\text{high}}$ can be obtained from Wang et al. (2012) and RH$_{\text{max}}$ is the RH at which $\gamma$ reaches $\gamma_{\text{high.}}$.

The rate of sulfate production via heterogeneous reactions $P_{\text{het(ox}_i)}$ can be expressed as:

$$P_{\text{het(ox}_i)} = \frac{3600 \times 96 \times p \times R_{\text{het(ox}_i)}}{RT}$$

,

(Eq. 16)

[Figure]

**Figure S3.** Plot of the SOR against aerosol water content (AWC) (note log scale), grouped by $O_3$ concentration. The solid blue circles represent $O_3 > 35$ ppb and the solid black circles represent $O_3 < 35$ ppb. The boxes represent, from top to bottom, the 75th, 50th, and 25th percentiles in each bin, which were also separated according to the 35 ppb $O_3$ concentration threshold; the bin widths were set such that there were an approximately equal number of data points in each bin. The whiskers, solid squares, and open circles represent 1.5 times the IQR, mean values, and outlier data points, respectively. The lines are best fits to the mean values based on a sigmoid function. Data for days with rain or snow were excluded from this plot.

[Figure]

**Figure S4.** Plots of $O_3$ against the primary emission tracers NO and $SO_2$.

[Figure]

**Figure S5.** Plots of sulfur oxidation ratios (SORs) against the primary emission tracers $SO_2$, NO, EC, and Se.

[Figure]

**Figure S6.** Time series of estimated $H_2O_2$ from March 1 2012 to February 28 2013. $H_2O_2$ was estimated from temperature (T) based on the fitting function $H_2O_2 = 0.1155e^{0.0846T}$ according to Fu (2014). The boxes represent, from top to bottom, the 75th, 50th, and 25th percentiles for each season. The whiskers, solid red squares,

and open red circles represent 1.5 times the interquartile range (IQR), seasonal mean values, and outlier

data points, respectively.

[Figure]

**Figure S7**. Plot of the SOR against estimated $H_2O_2$ grouped by RH. The solid blue circles represent RH > 45 % and

the solid black circles represent RH < 45 %. The boxes represent, from top to bottom, the 75th, 50th, and 25th

percentiles in each bin. The bin widths were set such that there were an approximately equal number of data points

in each bin. The whiskers, solid squares, and open circles represent 1.5 times the IQR, mean values, and outlier data

points, respectively. The line are best fits to the mean values based on an exponential function. Data for days with

rain were excluded from this plot.

[revised manuscript text omitted]